# Token Signature: Predicting Chain-of-Thought Gains with Token Decoding Feature in Large Language Models

**Peijie Liu** [1]  **Fengli Xu** [1]  **Yong Li** [1]

## Abstract

Chain-of-Thought (CoT) technique has proven effective in improving the performance of large language models (LLMs) on complex reasoning tasks. However, the performance gains are inconsistent across different tasks, and the underlying mechanism remains a long-standing research question. In this work, we make a preliminary observation that the monotonicity of token probability distributions may be correlated with the gains achieved through CoT reasoning. Leveraging this insight, we propose two indicators based on the token probability distribution to assess CoT effectiveness across different tasks. By combining instance-level indicators with logistic regression model, we introduce Dynamic CoT, a method that dynamically select between CoT and direct answer. Furthermore, we extend Dynamic CoT to closed-source models by transferring decision strategies learned from open-source models. Our indicators for assessing CoT effectiveness achieve an accuracy of 89.2%, and Dynamic CoT reduces token consumption by more than 35% while maintaining high accuracy. Overall, our work offers a novel perspective on the underlying mechanisms of CoT reasoning and provides a framework for its more efficient deployment. The code can be found at https://github.com/tsinghua-fib-lab/Token_Signature.

## 1. Introduction

Chain-of-Thought (CoT) prompting (Wei et al., 2022) has become a widely adopted technique for enhancing the reasoning capabilities of large language models(LLMs). By

incorporating examples of CoT reasoning in a few-shot prompt (Wei et al., 2022), CoT can be effectively triggered, and the ability of LLMs to solve various complex problems is improved by decomposing the problem step by step (Wang et al., 2022a), while also providing detailed and interpretable explanations (Lanham et al., 2023). Inspired by CoT reasoning, OpenAI (2024a) has introduced the concept of test-time scaling (Xu et al., 2025), which suggests that the reasoning capabilities of LLMs can be enhanced with more time spent thinking (test-time compute). For many problems such as mathematical word problems and symbolic reasoning, CoT is generally considered to be an effective method (Chae et al., 2024) (Qi et al., 2024).

However, recent studies have shown that CoT prompting does not consistently improve performance across all tasks, and its effectiveness varies depending on the problem domain. As illustrated in Figure 1, CoT's performance varies across different task categories. Moreover, in symbolic tasks, which are considered to be tasks where CoT is generally effective (Sprague et al., 2024), such as ContextHub-abductive and ContextHub-deductive (Hua et al., 2024), the significance of CoT gain also varies. In non-mathematical fields, for example, CoT has been found to be less effective (Kambhampati et al., 2024) and may even result in negative performance outcomes (Wang et al., 2024). Sprague et al. (2024) conducted a meta-analysis of CoT-related studies and experiments (Sprague et al., 2024), revealing that CoT is predominantly effective for mathematical and symbolic reasoning tasks, with limited or no improvements for other types of problems. However, this analysis (Sprague et al., 2024) focused solely on thematic trends and did not fully explore the underlying mechanisms driving CoT's effectiveness. While the effectiveness of CoT across different problems and models can be generally inferred from the task category, it remains inconsistent and lacks a definitive measure of effectiveness. Consequently, our work is motivated by two main goals: **to explore the underlying mechanisms of CoT reasoning** and **to develop a task-level method for evaluating its effectiveness**.

In this paper, we look at a novel perspective of the LLM decoding process. Instead of focusing on the most probable next token, we look at the token probability distribution and

[1]Department of Electronic Engineering, BNRist, Tsinghua University, China. Correspondence to: Fengli Xu <fenglixu@tsinghua.edu.cn>.

*Proceedings of the 42nd International Conference on Machine Learning*, Vancouver, Canada. PMLR 267, 2025. Copyright 2025 by the author(s).

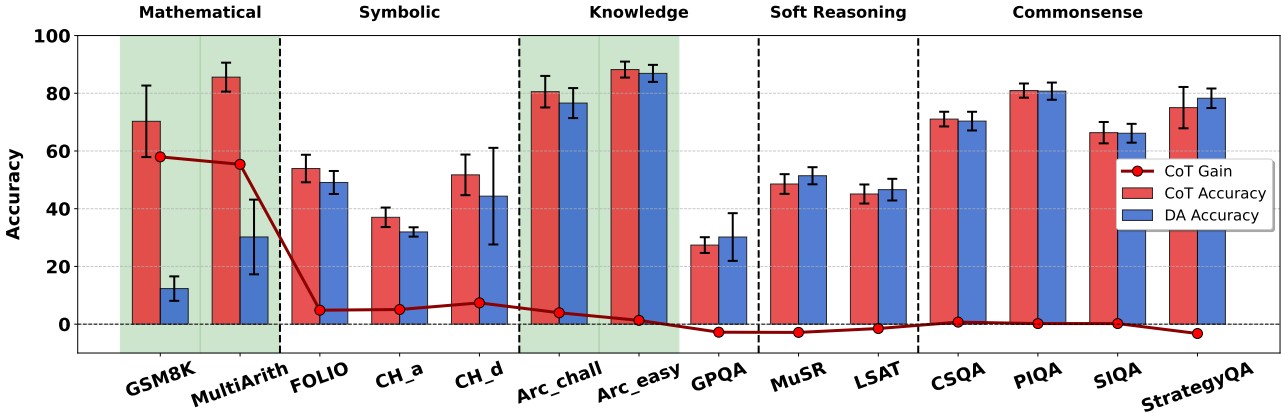

Figure 1: Average zero-shot CoT accuracy, direct answer (DA) accuracy, and CoT Gain across five benchmark categories for four open-source models. The benchmarks highlighted in green indicate that CoT is significantly greater than 0 (p<0.05). The categories include: Mathematical, Symbolic, Knowledge, Soft Reasoning, and Commonsense. The results show that CoT does not consistently lead to performance gain. Additionally, CoT's performance varies across different question types and task categories, and even within the same category, its effectiveness is inconsistent (with varying significance levels). This highlights that the utility of CoT cannot be solely determined by the question category.

how it changes as the number of tokens scales, which is what we defined as **Token Signature**. Specifically, we use standard prompts (i.e., questions only) to elicit responses and observe that the probability distribution of the initial token in the model's greedy decoding path is highly variable and closely correlated with CoT gain. Leveraging this insight, we develop two indicators based on the token probability distribution and Spearman Correlation (SC) (Wissler, 1905): **Instance SC** and **Aggregated SC**. These indicators quantify CoT effectiveness at the benchmark level. Secondly, we apply Instance-level SC to individual instances across different models and combine a small number of benchmark samples for classification, which allows us to dynamically select between CoT and direct answer. We refer to this approach as **Dynamic CoT**. Finally, we determine the best answer type (CoT or direct answer) by ensembling the results of a small open-source model at the question level, which is then transferred to a larger closed-source model for evaluation.

We test the effectiveness of our method on 12 well-known benchmarks, including GSM8K (Cobbe et al., 2021), Multi-Arith (Roy & Roth, 2016), CommonsenseQA (Talmor et al., 2018), and LSAT (Zhong et al., 2023), etc. We demonstrate its generalization across four closed-source models and two open-source models. At the benchmark level, we introduce two indicators that effectively predict the applicability of CoT. The positive and negative values of these indicators are closely correlated with CoT gains. Specifically, Instance SC achieves a prediction accuracy of 69.6%, while Aggregated SC reaches 89.2%. At the question level, the accuracy of our method, Dynamic CoT, is nearly identical to the highest

performance between CoT and direct answers. Compared to CoT, Dynamic CoT reduces token consumption by 39.1%. In the transfer experiment, our method still maintains high accuracy and reduces token consumption by 35.8%.

Our main contributions are as follows:

- We introduce the concept of **Token Signature** to study the CoT gain across different question types based on the decoded token probability distribution.

- We propose two token probability distribution indicators, **Instance SC** and **Aggregated SC**, to assess whether a benchmark is suitable for CoT at the benchmark level.

- We design **Dynamic CoT** at the instance level to enable the reasonable selection of either CoT or direct answer.

## 2. Preliminary Token-level Analysis

In this section, we present our initial observation on token probability distribution in open-source language models. We begin by conducting a token-level probability analysis using the publicly available Mistral-7B-Instruct model. The model is prompted with question-only inputs (i.e., without CoT trigger or direct answer trigger) and decoded using a greedy strategy. Figure 3 illustrates the probability distribution of the initially generated tokens across the four benchmarks.

Our preliminary result reveals distinct patterns across differ-

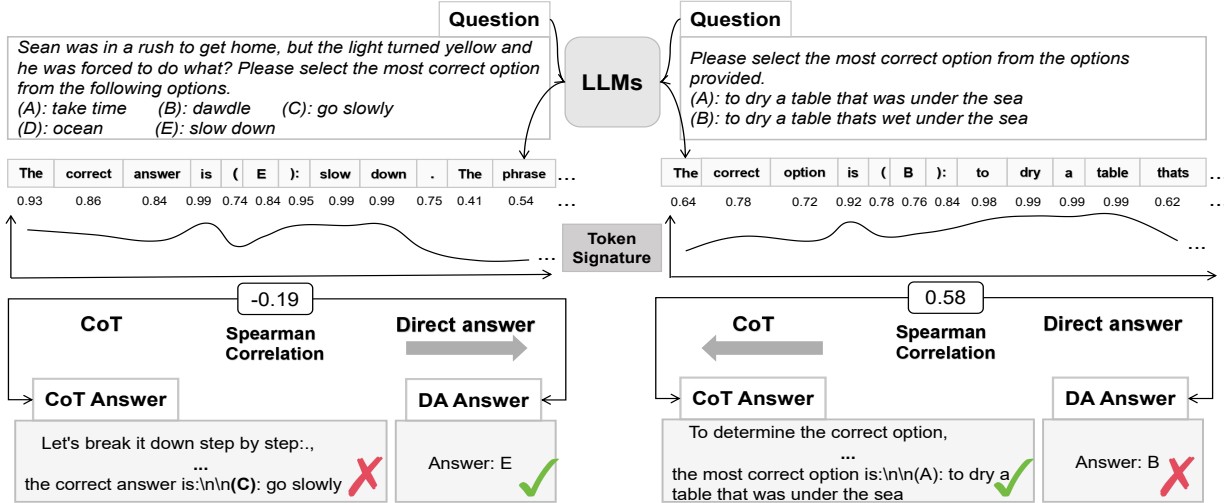

Figure 2: Illustration of the proposed method for analyzing CoT features through decoding. We calculate spearman correlation from token probability distribution obtained via greedy decoding of the standard prompt. This indicator reflects model confidence in answering a question and guides whether to introduce CoT reasoning after the standard prompt or to directly respond with a trigger prompt.

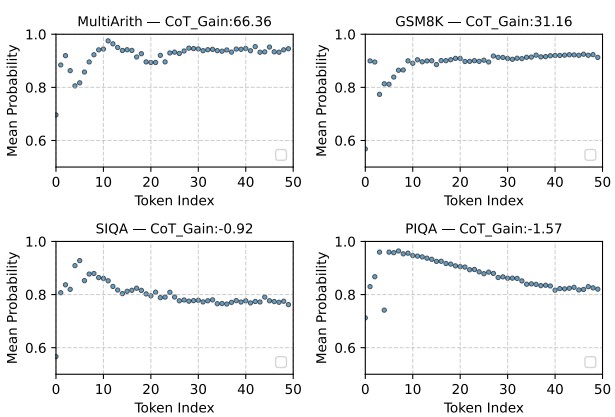

Figure 3: Probability distributions of the first 50 tokens generated along the trigger-free greedy decoding path for four benchmarks: MultiArith, GSM8K, SIQA and PIQA. The observed trends suggest a potential correlation between token probability distributions and CoT gain.

ent benchmarks. Notably, for benchmarks such as GSM8K and MultiArith, where CoT reasoning significantly enhances performance, the token probability distribution exhibits an increasing trend—indicating that later tokens are assigned higher probabilities, reflecting greater model confidence. In contrast, for benchmarks such as PIQA and SIQA, where CoT has almost no benefit, the probability distributions display a decreasing trend, suggesting a decline in model confidence as decoding progresses. Based on these observations, we propose the following hypothesis:

*The probability distribution of large language models along the decoding path is potentially correlated with the CoT gain across different question types.*

This insight motivates the approach introduced in the next section, where we leverage token probability distributions to characterize the features of CoT reasoning.

## 3. Our Approach

In this section, we introduce a novel perspective on the decoding process of LLMs. Rather than focusing solely on the most probable next token, we analyze the token probability distribution and its evolution over the decoding trajectory, which is what we define as **Token Signature**. We propose two key indicators to evaluate the effectiveness of CoT reasoning at the benchmark granularity. Next, we design an instance-granularity approach for dynamically selecting CoT. Finally, we develop a mechanism to adapt our method to closed-source models.

### 3.1. Token Signature

In Section 2, we preliminarily find that the trends of the probability distribution of tokens in different benchmarks are different. To capture this different feature, we introduce Spearman Correlation (Wissler, 1905) under standard prompt to measure the correlation between token probability and sequence order. A schematic diagram of the instance-level calculation of the token signature is shown in Figure 2.

For spearman correlation ($\rho_i$) (Wissler, 1905), that is, given two ranked variables $X = \{x_1, x_2, \ldots, x_n\}$ and

$Y = \{y_1, y_2, \ldots, y_n\}$, the $\rho_i$ is defined as:

$$\text{Spearman}(X, Y) = 1 - \frac{6 \sum d_i^2}{n(n^2 - 1)}, \quad (1)$$

where $d_i = R(x_i) - R(y_i)$ is the difference in ranks for each pair $(x_i, y_i)$, and $n$ is the number of observations.

**Instance SC** The Instance SC measures the monotonic relationship between the token probabilities and their sequence order within an individual response. For each question $q_i$, we extract the probability sequence of the first 50 tokens(typically covering 28% of the entire response):

$$P_i = \{p_{i,1}, p_{i,2}, \ldots, p_{i,50}\}, \quad (2)$$

where $p_{i,t}$ represents the model's softmax probability for the $t$-th token in the generated response to question $q_i$. We compute the spearman correlation between $P_i$ and its corresponding token index sequence $T = \{1, 2, \ldots, 50\}$:

$$\rho_i = \text{Spearman}(P_i, T). \quad (3)$$

Finally, the Instance SC is defined as the mean Spearman correlation across all test instances in a given benchmark:

$$\textbf{Instance SC} = \frac{1}{N} \sum_{i=1}^{N} \rho_i \quad (4)$$

where $N$ is the total number of questions in the benchmark.

**Aggregated SC** The Aggregated SC provides a benchmark-wide measure of token probability trends. Instead of computing Spearman Correlation per instance, we first compute the mean probability of each token index across all responses:

$$\bar{P}_t = \frac{1}{N} \sum_{i=1}^{N} p_{i,t}, \quad t \in \{1, 2, \ldots, 50\}, \quad (5)$$

where $\bar{P}_t$ represents the average probability assigned to the $t$-th token across all responses in the benchmark. We then compute the Spearman Correlation between the aggregated probability sequence $\bar{P} = \{\bar{P}_1, \bar{P}_2, \ldots, \bar{P}_{50}\}$ and the token index sequence $T = \{1, 2, \ldots, 50\}$:

$$\textbf{Aggregated SC} = \text{Spearman}(\bar{P}, T). \quad (6)$$

We use the two metrics, Instance SC and Aggregated SC, to predict the effectiveness of CoT on a specific benchmark. The significance of CoT is categorized into three levels: positive, none, and negative. The prediction results are determined as follows:

$$\textbf{Pred\_Significance} = \begin{cases} \text{positive,} & \text{if indicator} > 0, \\ \text{none/negative,} & \text{if indicator} \leq 0. \end{cases} \quad (7)$$

## 3.2. Dynamic CoT

We also propose a classification-based approach that leverages Instance-level SC to adaptively apply CoT reasoning. Given the variability in LLM's training data, we do not simply use zero as the threshold for instance-level classification. Instead, we introduce a logistic regression model trained on a small sample (50 instances) per benchmark, using instance-level SC as input and assigning labels. This trained model is then used to classify the remaining instances.

Specifically, the test label represents the better prompt to choose. When we selected the test set, we did not consider the case where CoT and DA had the same answer. For question $q_i$, $y_i$ be the binary label, where:

$$y_i = \begin{cases} 1, & \text{if answer of CoT is correct,} \\ 0, & \text{if answer of DA is correct.} \end{cases} \quad (8)$$

Combining the Instance-level SC and labels of equation (3), we train the logistic regression model:

$$P(y_i = 1 \mid \rho_i) = \frac{1}{1 + e^{-(w\rho_i + b)}}, \quad (9)$$

where $w$ and $b$ are the learned parameters, weight, and bias. We select the binary cross-entropy loss as a loss function:

$$L = -\frac{1}{N} \sum_{i=1}^{N} \left[ y_i \log \hat{y}_i + (1 - y_i) \log(1 - \hat{y}_i) \right], \quad (10)$$

During classification, if the predicted probability satisfies:

$$P(y_i = 1 \mid \rho_i) > 0.5, \quad (11)$$

then we classify $q_i$ as requiring CoT (*i.e.,* $y_i = 1$). Otherwise, the model generates a direct answer without CoT.

## 3.3. Transfer to Closed-source Model

Most closed-source models do not provide token probability outputs, making it challenging to apply our method. To address this limitation, we propose a voting mechanism to transfer our method.

We first evaluate the Dynamic CoT selection strategy in multiple open-source models to obtain multiple predicted $P_i$ for the question. We aggregate predictions from multiple open-source models using a voting mechanism. Specifically, the final label $Y_i$ for the instance $q_i$ is computed as:

$$Y_i = \mathbb{I}\left( \frac{1}{M} \sum_{m=1}^{M} P_i^{(m)} > 0.5 \right), \quad (12)$$

where $M$ is the total number of open-source models. $\mathbb{I}(\cdot)$ is the indicator function, which outputs 1 if the condition holds and 0 otherwise.

CoT is used only when $Y_i$ is 1, otherwise it is a direct answer. Based on the above voting results, the Dynamic CoT is then transferred to a closed-source model and then tested.

# 4. Experimental Setup

In this section, we will introduce the following aspects: model, benchmark, and prompt used in the experiment and how to evaluate the accuracy of the experiment.

## 4.1. Base Models

We conduct experiments primarily on four widely used open-source models and two popular closed-source models. The four open-source models include: Llama-3.2-3B-Instruct (Dubey et al., 2024), Phi-3.5-mini-instruct (Abdin et al., 2024), Llama-3.1-8B-Instruct (Dubey et al., 2024) and Mistral-7B-Instruct-v0.3 (Jiang et al., 2023). The two closed-source models are GPT-4o-mini and GPT-4o (OpenAI, 2024b).

## 4.2. Benchmark

For benchmarks, we refer to the categories outlined in Sprague et al. (2024)'s work. We focus on five types of benchmarks: Mathematical, Symbolic, Knowledge, Soft Reasoning, and Commonsense. The benchmarks used are categorized as follows:

- **Mathematical**: GSM8K (Cobbe et al., 2021), Multi-Arith (Roy & Roth, 2016)

- **Symbolic**: FOLIO (Han et al., 2022), ContextHub (Hua et al., 2024)

- **Knowledge**: ARC (Clark et al., 2018), GPQA (Rein et al., 2023)

- **Soft Reasoning**: MuSR (Sprague et al., 2023), AGIEval LSAT (Zhong et al., 2023)

- **Commonsense**: CommonsenseQA (Talmor et al., 2018), PIQA (Bisk et al., 2020), SIQA (Sap et al., 2019), StrategyQA (Geva et al., 2021)

The answer formats mainly include two types: short-answer and multiple-choice. We introduce the specific benchmark details in the Appendix A.

## 4.3. Prompt Settings

We utilize two primary types of prompts: zero-shot CoT prompt (Kojima et al., 2022), and zero-shot direct answer(DA) prompt. For the zero-shot CoT prompt, we employ the phrase "Let's think step by step" (Kojima et al., 2022) as the CoT trigger. Additionally, we designed a direct answer prompt tailored to different benchmarks to ensure the model adhered to the provided instructions. Detailed descriptions of the prompts can be found in the Appendix A.

## 4.4. Evaluation

**Answer Extract** To extract answers from the model's response, we employ distinct strategies tailored to different question types. For short-answer mathematical reasoning questions, we select the final numerical value in the model's output as the answer, adhering to a widely accepted protocol for evaluating language models (Ivison et al., 2023; Wang et al., 2023). For multiple-choice questions, we identify the first letter of the option provided in the direct response as the answer. In the case of the CoT response, we append the prompt "So the best answer letter choice is" to extend the response, subsequently extracting the corresponding letter option as the answer, and then matching them with the correct answer.

**Answer Accuracy** We evaluate the accuracy of the answers by comparing the extracted answers with the correct ones. For each evaluation, we calculate the accuracy as:

$$\text{Acc} = N_{\text{correct}}/N, \tag{13}$$

where $N_{\text{correct}}$ denotes the number of correctly answered questions, and $N$ represents the total number of questions.

**Significance Judgment** To assess the significance of CoT gain on a benchmark, we perform a two-tailed Z test. The null hypothesis assumes no significant difference between DA Acc ($p_1$) and CoT Acc ($p_2$), i.e., $p_1 = p_2$. The alternative hypothesis tests whether the difference $p_2 - p_1$ is significantly different, i.e., $p_2 \neq p_1$. The detailed calculation process is provided in Appendix A.

# 5. Results

## 5.1. CoT Effectiveness at the Benchmark Level

In this section, we present the results of using two Token Signature indicators(Instance SC and Aggregated SC)to predict the effectiveness of CoT reasoning across different benchmarks.

We evaluate CoT reasoning and direct answer performance on 12 benchmarks using 4 open-source models, with all results summarized in Table 7. Additionally, we compute the values of Instance SC and Aggregated SC for each benchmark and record their corresponding CoT gain in Table 1. Table 1 presents the results of the Llama-3.2-3B-Instruct model. The values of Instance SC and Aggregated SC exhibit a strong correlation with CoT gain, where their signs (positive or negative) align closely with the effectiveness of CoT reasoning. For benchmarks with significant CoT gains, such as GSM8K, MultiArith, FOLIO, and CH_d, both Instance SC and Aggregated SC are positive. Conversely, for benchmarks with minimal or negative CoT gains, such as MuSR, LSAT, SIQA, and StrategyQA, both indicators are negative. A similar trend can be observed in Table 8.

Table 1: Instance SC, Aggregated SC, and CoT Gain across benchmarks on Llama-3.2-3B-Instruct. SC indicators (Instance SC/Aggregated SC) effectively predict CoT effectiveness at the benchmark granularity. Significance indicates the significance of CoT gain determined using the Z test (positive/none/negative). For more complete information, see Table 7 and 8.

|  | Instance SC | Aggregated SC | CoT Gain | Significance |
|---|---|---|---|---|
| GSM8K | 0.0450 | 0.2080 | 63.76 | positive |
| MultiArith | 0.1619 | 0.2904 | 67.83 | positive |
| FOLIO | 0.1869 | 0.0488 | 6.31 | positive |
| CH_a | 0.0720 | -0.032 | 1.67 | none |
| CH_d | 0.0900 | 0.2670 | 19.95 | positive |
| Arc_chall | -0.0513 | -0.4597 | 4.01 | positive |
| Arc_easy | -0.0552 | -0.6061 | 0.93 | none |
| GPQA | 0.0038 | -0.1042 | -10.71 | negative |
| MuSR | -0.0840 | -0.4016 | -6.09 | negative |
| LSAT | -0.0661 | -0.4180 | -0.99 | none |
| CSQA | -0.1102 | -0.3424 | 1.64 | none |
| PIQA | -0.2698 | -0.5340 | 1.74 | none |
| SIQA | -0.2698 | -0.7639 | 0.36 | none |
| StrategyQA | -0.0082 | -0.3917 | -13.1 | negative |

We hypothesize that when these indicators are negative during the model's reasoning process, it suggests a high degree of uncertainty, which leads to the inadaptability of CoT reasoning. This implies that CoT is less effective in scenarios where token probability distributions indicate lower confidence in sequential reasoning.

Notably, for symbolic benchmarks such as CH_d, CoT exhibits a negative effect on Mistral-7B-Instruct-v0.3 and Phi-3.5-mini-instruct, with their corresponding Aggregated SC values also being negative. Conversely, CoT has a positive effect on Llama-3.2-3B-Instruct and Llama-3.1-8B-Instruct, where their Aggregated SC values are positive.

We further evaluate the accuracy of Instance SC and Aggregated SC in predicting the effectiveness of CoT across four models. To classify the impact of CoT, we define three categories using Z test CoT gain: positive significance (CoT improves performance), no significance (CoT has negligible impact), and negative significance (CoT reduces performance). The prediction is considered accurate if the indicator is greater than 0 when CoT has a positive gain, or less than 0 when CoT has no gain or a negative gain, which indicates that CoT is ineffective or even harmful. The detailed accuracy results are reported in Table 2.

We observe that both indicators demonstrate high accuracy in predicting CoT effectiveness. On average, Instance SC achieves 69.6% accuracy, while Aggregated SC performs even better, reaching 89.2% accuracy. These results further

Table 2: Prediction accuracy of two indicators for zero-shot CoT effectiveness across four open-source models.

|  | Instance SC | Aggregated SC |
|---|---|---|
| Llama-3.2-3B | 78.6 | 92.9 |
| Mistral-7B | 50.0 | 85.7 |
| Phi-3.5-mini | 78.6 | 85.7 |
| Llama-3.1-8B | 71.4 | 92.9 |

reinforce the potential correlation between token probability distribution and CoT gain, while also validating the predictive power of our proposed indicators. Compared with Instance SC, Aggregated SC has better prediction ability, which may be attributed to the divergence of instance-granular token probability distribution.

**5.2. Dynamic CoT**

In Section 3, we introduce our Dynamic CoT, which integrates Instance-level SC with a logistic regression model to explore the impact of question granularity and dynamically select CoT and DA. The experimental results for Dynamic CoT across four open-source models are presented in Table 9, while the average performance of Dynamic CoT on these models is summarized in Table 3.

Table 3: Average performance impact of Dynamic CoT across benchmarks on four open-source models. Gap denotes the relative difference between Dynamic CoT and the better of CoT and DA. Dynamic CoT consistently achieves the higher accuracy between CoT and direct answers on most benchmarks.

|  | CoT Acc | DA Acc | Dynamic CoT | Gap |
|---|---|---|---|---|
| GSM8K | **70.28** | 12.32 | 69.29 | -1.4 |
| MultiArith | **85.59** | 30.21 | 85.00 | -1.0 |
| FOLIO | 53.93 | 49.09 | **54.05** | 0.2 |
| CH_a | **37.02** | 31.95 | 35.29 | -4.7 |
| CH_d | 51.73 | 44.20 | **53.06** | 3.6 |
| Arc_chall | 80.55 | 76.60 | **81.20** | 0.8 |
| Arc_easy | 88.21 | 86.89 | **88.65** | 0.5 |
| GPQA | 27.40 | **30.19** | 28.52 | -5.5 |
| MuSR | 48.81 | **51.42** | 50.99 | -0.8 |
| LSAT | 45.10 | **46.61** | 45.99 | -1.3 |
| CSQA | 71.05 | 70.35 | **71.54** | 1.7 |
| PIQA | 80.93 | 80.73 | **82.31** | 1.7 |
| SIQA | 66.35 | 66.15 | **66.89** | 0.8 |
| StrategyQA | 75.02 | 78.27 | **79.54** | 1.6 |

In Table 9 and 10, we can see that the performance of Dynamic CoT basically reaches the highest performance among all CoT and all direct answers. Specifically, in a series of 56 experimental setups across four models, Dynamic CoT achieves the highest performance in 46.4% of

the cases within CoT, direct answer, and Dynamic CoT comparison, and ranks among the top two methods in 92.8% of these experiments. Additionally, Table 3 demonstrates that Dynamic CoT achieve the best average performance across eight benchmarks and the second-best performance across six others. These results demonstrate that our method is highly effective at achieving classification-level performance on a question-specific granularity.

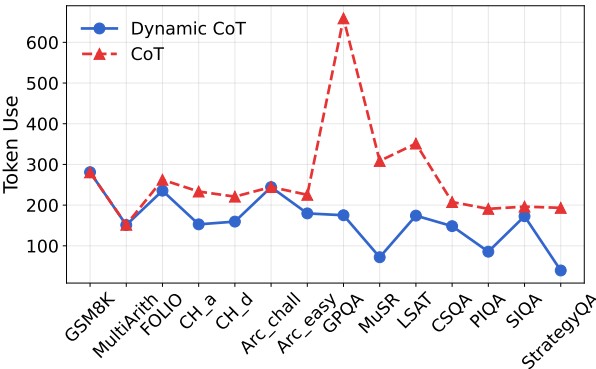

Figure 4: Comparison of token consumption between Dynamic CoT and All CoT in multiple open-source models of multiple benchmarks

Furthermore, Figure 4 presents the average token consumption across 14 benchmarks for both CoT and Dynamic CoT. The findings indicate that Dynamic CoT uses fewer tokens than CoT on eight benchmarks, with significant reductions observed in GPQA, MuSR, PIQA, and StrategyQA. Overall, Dynamic CoT reduces token consumption by an average of 104 tokens, representing a 39.1% reduction compared to CoT.

In summary, our Dynamic CoT method has been effectively applied to open-source models, enabling dynamic selection between CoT and direct answers while maintaining high accuracy. Furthermore, our approach significantly reduces unnecessary token generation compared to the direct use of CoT, particularly in benchmarks where CoT is less effective.

### 5.3. Model Transfer

For LLMs that cannot be deployed locally, the token probability distribution may be inaccessible, and the evaluation cost tends to be relatively high. As a result, a transfer strategy is required to adapt our method to closed-source models. So we propose an approach to integrate results from smaller open-source models, which are then applied to the closed-source model. The details of this method are outlined in Section 3.

Table 11 presents the experimental results for each benchmark on GPT-4o-mini and GPT-4o. The results in Table 4 indicate that Dynamic CoT performs well, ranking first

Table 4: The cross-model transfer effect of Dynamic CoT from open-source to closed-source models across benchmarks. Gap denotes the relative difference between Dynamic CoT and the better of CoT and DA. Experimental results demonstrate significant performance improvements on closed-source models and multiple benchmarks.

| | CoT Acc | DA Acc | Dynamic CoT | Gap |
|---|---|---|---|---|
| GSM8K | **84.76** | 42.46 | **84.76** | 0.0 |
| MultiArith | 93.92 | **96.42** | 93.92 | -0.5 |
| FOLIO | **72.14** | 65.91 | **72.14** | 0.0 |
| CH_a | **58.17** | 44.19 | **58.17** | 0.0 |
| CH_d | **57.17** | 48.34 | **57.17** | 0.0 |
| Arc_chall | **94.11** | 92.19 | **94.11** | 0.0 |
| Arc_easy | **94.30** | 94.28 | **94.30** | 0.0 |
| GPQA | **51.68** | 48.11 | 47.88 | -7.4 |
| MuSR | **60.78** | 57.61 | 57.61 | -5.2 |
| LSAT | **68.93** | 68.34 | 68.59 | -0.5 |
| CSQA | 82.76 | **83.29** | 82.72 | -0.7 |
| PIQA | 89.97 | **91.70** | 90.29 | -1.5 |
| SIQA | **77.49** | 77.05 | 77.46 | 0.0 |
| StrategyQA | **61.77** | 54.72 | 54.76 | -11.3 |

or second in the majority of experiments across the three conditions. As shown in Figures 4 and 5, compared to the native Dynamic CoT, our method exhibits a slight decrease in classification performance. In benchmarks such as GSM8K, MultiArith, and FOLIO, Dynamic CoT exhibits performance equivalent to CoT. This outcome arises from the voting process, where the method ultimately selects CoT for these benchmarks. Notably, these benchmarks show relatively high gains from CoT, highlighting that our approach effectively identifies when CoT is appropriate at the benchmark level.

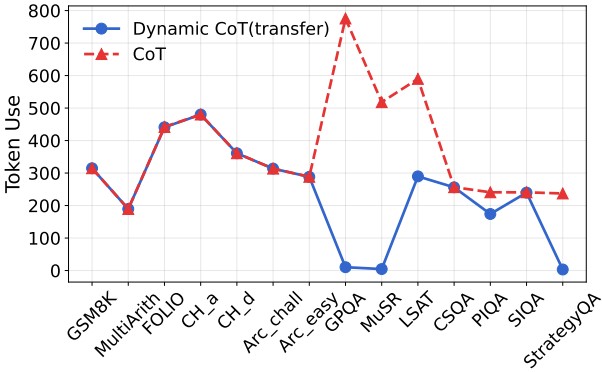

Figure 5: Average token consumption of the Dynamic CoT approach after transferring the approach from open-source to closed-source. A comparison of token consumption between Dynamic CoT (transfer) and All CoT across multiple models and benchmarks.

Figure 5 presents the token consumption for each bench-

mark on the GPT model. Results show no change in token consumption for benchmarks such as GSM8K, MultiArith, and FOLIO, while a decrease is observed for benchmarks like GPQA, MuSR, and LSAT. On average, our method consumes 134 fewer tokens than all CoT methods, resulting in a 35.8% reduction.

Overall, after transferring using the voting method, Dynamic CoT retains high accuracy, although its fine-grained classification ability experiences a slight reduction. Additionally, this approach consumes fewer tokens compared to all CoT methods.

## 6. Analysis

### 6.1. Impact of SC Threshold

To examine the effect of token count on SC-based prediction performance, we conduct experiments using the top n tokens, where $n \in \{10, 20, 50, 100, 200\}$. We compare the accuracy of both Instance SC and Aggregated SC under these settings. The results are presented in Figure 6. Notably, using the first 50 tokens yields the highest prediction performance, 69.6% for Instance SC and 89.3% for Aggregated SC.

Based on these findings, we empirically select 50 as the default token threshold in our experiments. We observe that smaller n values fail to capture sufficient correlation trends, while larger values suffer from sparsity issues due to shorter decoding paths in many samples. Therefore, using 50 tokens provides a balanced trade-off between signal strength and data availability.

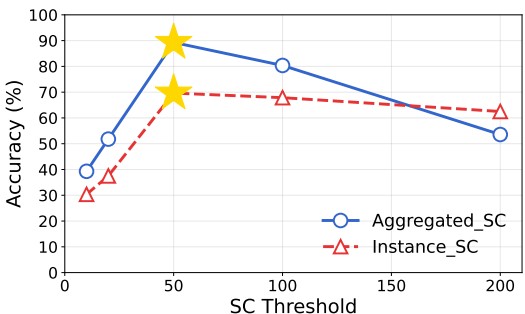

Figure 6: Prediction accuracy of Instance SC and Aggregated SC under varying token thresholds. The best performance is achieved when using the top 50 tokens.

### 6.2. Impact of Decoding Strategies

Decoding strategy is a critical factor influencing the outputs of large language models. We systematically evaluate the impact of different decoding strategies, focusing primarily on temperature and Top-K sampling, using the Llama-3.2-3B-Instruct model. We perform a sensitivity analysis across various samples and compare the prediction accuracy of

two CoT effectiveness indicators at the benchmark granularity under these varying decoding configurations. The results, presented in Table 12, indicate that our method consistently maintains strong predictive power and robustness despite changes in decoding strategies. Notably, across different task types, Token Signature exhibits similar predictive characteristics for assessing the effectiveness of chain-of-thought reasoning.

### 6.3. Impact of CoT Prompt

We investigate the impact of alternative prompt settings, such as few-shot CoT prompting. We conduct experiments using two indicators, with detailed results presented in Table 13. The prediction rates for each indicator are statistically summarized in Table 5. The experimental results show that our method still has good performance under few-shot CoT. Furthermore, Token Signature effectively predicts the gains achieved by few-shot CoT. Overall, the results demonstrate strong robustness.

Table 5: Prediction accuracy of Instance SC and Aggregated SC for few-shot CoT effectiveness across four open-source models.

|  | Instance SC | Aggregated SC |
|---|---|---|
| Llama-3.2-3B | 92.9 | 92.9 |
| Mistral-7B | 50.0 | 85.7 |
| Phi-3.5-mini | 57.1 | 78.6 |
| Llama-3.1-8B | 71.4 | 92.9 |

### 6.4. Intuitive Theoretical Analysis

Token probability has been shown to reflect a language model's confidence in its outputs (Farquhar et al., 2024). CoT prompting can improve performance on tasks with inherently sequential structures by enabling a deeper, token-expensive search process (Li et al., 2024). However, for tasks that are not intrinsically sequential, CoT may have adverse effects due to the accumulation of reasoning errors, also known as the snowball effect (Gan et al., 2025).

Mathematical reasoning tasks, such as those found in GSM8K, require a strict step-by-step process, where operations (e.g., arithmetic calculations, logical deductions) must be executed in a well-defined order. The solution space for such problems is highly constrained, and the generation of intermediate steps must adhere to deterministic rules. In this context, CoT acts as a structured reasoning scaffold, effectively enhancing the model's confidence and improving solution accuracy.

Conversely, in tasks such as commonsense reasoning, the solution space is more diverse and less rule-governed. When the model's initial confidence is low, CoT may lead it down

an incorrect reasoning path, with each subsequent step compounding earlier errors. This error propagation amplified by CoT can degrade performance.

**Token Signature as Early Predictor** We leverage the Spearman Correlation indicators between token probabilities and correct reasoning paths as a proxy for the model's uncertainty at the onset of the reasoning process. A high SC indicates that the model's internal token confidence is well-aligned with successful reasoning trajectories, suggesting that CoT is likely to yield performance gains. Thus, SC serves as an effective indicator for predicting the utility of CoT across different task types.

# 7. Related Work

## 7.1. Chain-of-Thought Reasoning

CoT reasoning enhances LLMs by generating intermediate reasoning steps, thereby improving both interpretability and performance on complex tasks. Few-shot CoT, first introduced by (Wei et al., 2022), enables CoT reasoning with only a few examples, significantly boosting performance in tasks such as arithmetic and symbolic reasoning. Kojima et al. (2022) proposed zero-shot CoT, which initiates the reasoning process using the prompt "Let's think step by step", enabling CoT reasoning without requiring labeled examples. In addition, numerous CoT variants have emerged, including Auto-CoT (Zhang et al., 2022), ToT (Yao et al., 2024), and Coconut (Hao et al., 2024), each designed to further enhance the generalization and effectiveness of CoT reasoning across various domains. In addition, CoT is also widely used (Shao et al., 2024; Shang et al., 2024b; Chen et al., 2024; Meng et al., 2025).

However, recent studies have demonstrated that CoT reasoning exhibits inconsistent performance across different benchmarks (Sprague et al., 2024). While CoT significantly enhances mathematical and symbolic reasoning tasks, its impact on commonsense reasoning and factual question answering remains limited (Kojima et al., 2022). CoT is particularly effective for structured, decomposable tasks but offers minimal improvement in tasks that require external knowledge or lack explicit reasoning steps. Furthermore, Liu et al. (2024) argues that CoT should not be indiscriminately applied to all tasks, as it significantly degrades the performance of LLMs in scenarios where excessive reasoning-akin to overthinking-detrimentally affects human performance. Although various techniques have been proposed to mitigate CoT's instability across tasks, such as Self-Consistency (Wang et al., 2022b), Program-of-Thought (Chen et al., 2022), Division-of-Thoughts (Shao et al., 2025), and Synergy-of-Thoughts (Shang et al., 2024a) the underlying principles governing CoT remain an open research question actively explored by the community.

## 7.2. Decoding for Large Language Models

The decoding process plays a crucial role in LLMs. Popular algorithms such as greedy decoding, temperature sampling (Ficler & Goldberg, 2017), top-k sampling (Radford et al., 2019), and diverse beam search (Vijayakumar et al., 2018) are often employed to enhance the quality of generated responses. In addition, Li et al. (2022) proposed a new decoding strategy called Contrastive Decoding to enhance output quality in open-ended text generation tasks. Shi et al. (2023) introduced Context-Aware Decoding, which focuses on reducing hallucination during the generation process. Recent study (Wang & Zhou, 2024) has shown that the CoT path can be spontaneously generated in the decoding path. By leveraging the information in the decoding process, the high confidence of the answer produced by the model can be used to find the CoT path without explicit prompt words (Wang & Zhou, 2024). While most existing studies primarily rely on next-token prediction, there has been limited exploration from the perspective of the overall token probability distribution. This gap is addressed in our work.

# 8. Conclusion

In this paper, we propose a novel perspective on LLM decoding by focusing on the token probability distribution rather than the most probable next token to analyze CoT reasoning. We introduce the concept of **Token Signature** to explore the correlation between the token probability distribution and CoT gains. Based on this insight, we develop two metrics, **Instance SC** and **Aggregated SC**, which effectively quantify CoT effectiveness at both the instance and benchmark levels. We further design **Dynamic CoT**, combining instance-level SC, to dynamically select between CoT and direct answers.

Extensive experiments across 12 widely used benchmarks validate the effectiveness of our approach on both open-source and closed-source models. At the benchmark level, our metrics predict the applicability of CoT with high accuracy, achieving 69.6% for **Instance SC** and 89.2% for **Aggregated SC**. At the question level, **Dynamic CoT** achieves performance comparable to the best results between CoT and direct answers while reducing token consumption by 39.1%. In transfer experiments, we observe that **Dynamic CoT** maintains high accuracy and further reduces token consumption by 35.8%.

Overall, we leverage token signature to assess CoT effectiveness and introduce a powerful mechanism for dynamically selecting the most effective answer strategy, while minimizing computational cost. Future work can explore extending the token signature concept, providing deeper insights into the CoT mechanism, and contributing to the development of more efficient large language models.

## Acknowledgements

This work was supported in part by the National Natural Science Foundation of China under 23IAA02114, 62472241, and Beijing National Research Center for Information Science and Technology.

## Impact Statement

This paper presents work whose goal is to advance the field of Machine Learning. There are many potential societal consequences of our work, none which we feel must be specifically highlighted here.

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

# A. Implementation Details

**Benchmark**   We use 12 widely used benchmarks, which are described in Table 6. Among them, We use the abductive and deductive data of level = 2 in ContextHub, represented by CH_a and CH_b. We use the challenge and easy data in ARC, represented by Arc_chall and Arc_easy.

Table 6: Introduction to the benchmark used in the paper.

|  | Category | Answer Format | Number | Brief Description |
|---|---|---|---|---|
| GSM8K(Cobbe et al., 2021) | Mathematical | Short Answer | 1319 | A dataset containing high-quality and diverse elementary school math word problems, designed to evaluate the mathematical reasoning ability in the model. |
| MultiArith(Roy & Roth, 2016) | Mathematical | Short Answer | 600 | A benchmark focused on multi-step arithmetic reasoning tasks that require models to solve math word problems involving basic operations. |
| FOLIO(Han et al., 2022) | Symbolic | True/False | 1204 | A dataset designed to test models on reasoning with first-order logic statements and deriving logical conclusions. |
| CH_a(Hua et al., 2024) | Symbolic | True/False | 2400 | A benchmark for testing abductive reasoning, where models must infer the most plausible explanation given a scenario. |
| CH_d(Hua et al., 2024) | Symbolic | True/False | 2400 | A benchmark focused on deductive reasoning tasks, requiring models to derive conclusions logically based on premises. |
| Arc_chall(Clark et al., 2018) | Knowledge | Multiple choice | 1172 | A challenging subset of the AI2 Reasoning Challenge (ARC) that contains difficult science questions for models to solve. |
| Arc_easy(Clark et al., 2018) | Knowledge | Multiple choice | 2376 | An easier subset of the AI2 Reasoning Challenge (ARC) designed to evaluate basic science knowledge and reasoning. |
| GPQA(Rein et al., 2023) | Knowledge | Multiple choice | 448 | The General Physics Question Answering benchmark assesses a model's ability to answer questions about fundamental physics concepts. |
| MuSR (Sprague et al., 2023) | Soft Reasoning | Multiple choice | 756 | A benchmark for evaluating Multistep Symbolic Reasoning, where models must solve tasks involving symbolic manipulation. |
| LSAT(Zhong et al., 2023) | Soft Reasoning | Multiple choice | 1009 | Based on the Law School Admission Test, this benchmark tests logical reasoning and reading comprehension. |
| CSQA(Talmor et al., 2018) | Commonsense | Multiple choice | 1221 | A benchmark to evaluate models' commonsense reasoning by answering questions that require real-world understanding. |
| PIQA(Bisk et al., 2020) | Commonsense | Multiple choice | 1838 | The Physical Interaction Question Answering benchmark focuses on models' understanding of everyday physical commonsense. |
| SIQA (Sap et al., 2019) | Commonsense | Multiple choice | 1954 | Social Interaction Question Answering, testing models' ability to reason about social situations and motivations. |
| StrategyQA(Geva et al., 2021) | Commonsense | True/False | 1508 | A benchmark designed for multi-step reasoning tasks where models must strategically reason to answer open-ended questions. |

**Detailed Prompt Setting**   For standard prompts, we carefully craft prompts for all benchmarks. For benchmarks with a True/False answer format, we restructure them as multiple-choice questions. For Zero-shot CoT prompts, we use the phrase "Let's think step by step" (Kojima et al., 2022) to trigger the CoT reasoning process. For direct answer prompts, we meticulously design them to ensure the model adheres to instructions. For benchmarks requiring short answers, we use the directive: "Your answer must not include any reasoning step. You must only write your numerical answer directly. You only output 'The answer is <answer>' where <answer> is the numerical answer to the problem," as the DA trigger. For multiple-choice benchmarks, we employ: "Your answer must not include any reasoning. You must write your answer directly. Write the answer in the following format: 'Answer: <Your Answer Letter Choice>'" as the DA trigger.

The open-source models utilized in our work are all instruction fine-tuned, and therefore, we employ different placeholders to encapsulate prompt tokens for each model. For Llama-3.2-3B-Instruct and Llama-3.1-8B-Instruct (Dubey et al., 2024), we adopt '<|begin_of_text|><|start_header_id|>user<|end_header_id|>' and '<|eot_id|><|start_header_id|>assistant<|end_header_id|>'. For Phi-3.5-mini-instruct (Abdin et al., 2024), we adopt '<|user|>' and '<|end|><|assistant|>'. For Mistral-7B-Instruct-v0.3 (Jiang et al., 2023), we adopt '[INST]' and '[/INST]'.

**Detailed Experimental Parameter Setting** We aim to maintain consistent experimental parameters across different models for the same benchmark. In general, we set do_sample = False or temperature = 0 to ensure greedy decoding. For standard prompt experiments, the maximum number of generated tokens is set to 50 (max_tokens=50); for CoT experiments, it is set to 1024 (max_tokens=1024); and for direct answer experiments, it is set to 32 (max_tokens=32). Minor adjustments are made in some experiments as needed.

**Significance Judgment** We use a two-tailed Z test to assess the significance of the difference between CoT Acc and DA Acc. For a specific benchmark, let $p_1$ and $p_2$ represent the accuracy rates under DA and CoT, respectively, and let $n_1$ and $n_2$ represent the sample sizes, both of which are equal to $N$, as detailed in the Table 6. The null hypothesis assumes no significant difference between the two values, i.e., $p_1 = p_2$. The alternative hypothesis tests whether the difference $p_2 - p_1$ is significantly different, i.e., $p_2 \neq p_1$. The Z test statistic is calculated using the following formula:

$$Z = \frac{(p_2 - p_1)}{\sqrt{\frac{p_0(1-p_0)}{n_1} + \frac{p_0(1-p_0)}{n_2}}},$$

where $p_0$ is the pooled proportion, computed as:

$$p_0 = \frac{p_1 n_1 + p_2 n_2}{n_1 + n_2}.$$

The p-value for a two-sided test is calculated as follows:

$$p = 2 \cdot P(z > |Z|).$$

Under the null hypothesis, the test statistic $Z$ follows a standard normal distribution. If the absolute value of the calculated Z-score exceeds the critical value associated with the desired significance level (typically 1.96 for a 95% confidence level) or if the p-value is less than 0.05, we reject the null hypothesis. This indicates that the observed difference is statistically significant. Specifically, if $|Z|$ surpasses the threshold for a 95% confidence level or $p < 0.05$, we conclude that $p_2$ represents a statistically significant change from $p_1$. Conversely, if these conditions are not met, we fail to reject the null hypothesis, suggesting that there is no significant improvement in CoT. Specifically, the significance of the result is determined as follows:

$$\textbf{Significance} = \begin{cases} \text{positive}, & \text{if } Z > 0 \text{ and } p < 0.05, \\ \text{none}, & \text{if } Z = 0 \text{ or } p \geq 0.05, \\ \text{negative}, & \text{if } Z < 0 \text{ and } p < 0.05. \end{cases}$$

**Compute Resources** We deploy four open-source models for inference on an A100 GPU with 80 GB RAM. Each experiment for each benchmark takes from a few minutes to a few hours, depending on the number of questions and the experiment type (Standard/CoT/Direct answer). For experiments on closed-source models, we use the official API interface of OpenAI [1].

# B. Supplementary experimental results

This section presents the supplementary experimental results from the paper. Specifically, Table 7 provides the CoT accuracy, direct answer accuracy, CoT gain, and significance judgment across 12 benchmarks for 4 open-source models and 2 closed-source models. Table 8 displays the experimental results for Instance SC and Aggregated SC on three models not covered in the main text. Table 9 and Table 10 outline the performance of Dynamic CoT on four models, comparing it to both CoT and direct answer approaches. Finally, Table 11 details the transfer of Dynamic CoT to a closed-source model and compares its performance with CoT and direct answer.

---

[1] https://openai.com/index/openai-api/

Table 7: Zero-shot CoT, direct answer accuracy and significance judgment on different benchmarks and different models

| | Model | CoT Acc | DA Acc | CoT Gain | Z Statistic | p Value | Significance |
|---|---|---|---|---|---|---|---|
| GSM8K | Llama-3.2-3B-Instruct | 72.10 | 8.34 | 63.76 | 33.39 | 0.000 | positive |
| | Mistral-7B-Instruct-v0.3 | 49.58 | 8.34 | 41.24 | 23.35 | 0.000 | positive |
| | Phi-3.5-mini-instruct | 78.24 | 18.35 | 59.89 | 30.78 | 0.000 | positive |
| | Llama-3.1-8B-Instruct | 81.20 | 14.25 | 66.95 | 34.42 | 0.000 | positive |
| | GPT-4o-mini | 84.38 | 31.01 | 53.37 | 27.74 | 0.000 | positive |
| | GPT-4o | 85.14 | 53.90 | 64.83 | 17.43 | 0.000 | positive |
| MultiArith | Llama-3.2-3B-Instruct | 88.50 | 20.67 | 67.83 | 23.60 | 0.000 | positive |
| | Mistral-7B-Instruct-v0.3 | 77.67 | 14.33 | 63.34 | 22.01 | 0.000 | positive |
| | Phi-3.5-mini-instruct | 85.17 | 44.33 | 40.84 | 14.81 | 0.000 | positive |
| | Llama-3.1-8B-Instruct | 91.00 | 41.50 | 49.50 | 18.13 | 0.000 | positive |
| | GPT-4o-mini | 94.33 | 94.83 | -0.5 | -0.38 | 0.702 | none |
| | GPT-4o | 93.50 | 98.00 | -4.5 | -3.86 | 0.000 | negative |
| FOLIO | Llama-3.2-3B-Instruct | 48.92 | 42.61 | 6.31 | 3.11 | 0.002 | positive |
| | Mistral-7B-Instruct-v0.3 | 49.42 | 51.00 | -1.58 | -0.78 | 0.438 | none |
| | Phi-3.5-mini-instruct | 59.14 | 53.32 | 5.82 | 2.88 | 0.004 | positive |
| | Llama-3.1-8B-Instruct | 58.22 | 49.42 | 8.80 | 4.33 | 0.000 | positive |
| | GPT-4o-mini | 69.19 | 62.21 | 6.98 | 3.61 | 0.000 | positive |
| | GPT-4o | 75.08 | 69.60 | 5.48 | 3.00 | 0.003 | positive |
| CH_a | Llama-3.2-3B-Instruct | 35.21 | 33.54 | 1.67 | 1.22 | 0.223 | none |
| | Mistral-7B-Instruct-v0.3 | 32.37 | 30.17 | 2.20 | 1.64 | 0.100 | none |
| | Phi-3.5-mini-instruct | 40.29 | 33.58 | 6.71 | 4.82 | 0.000 | positive |
| | Llama-3.1-8B-Instruct | 40.21 | 30.50 | 9.71 | 7.03 | 0.000 | positive |
| | GPT-4o-mini | 56.92 | 46.25 | 10.67 | 7.40 | 0.000 | positive |
| | GPT-4o | 59.42 | 42.13 | 17.29 | 11.98 | 0.000 | positive |
| CH_d | Llama-3.2-3B-Instruct | 39.87 | 19.92 | 19.95 | 15.10 | 0.0 | positive |
| | Mistral-7B-Instruct-v0.3 | 55.92 | 59.04 | -3.12 | -2.19 | 0.029 | negative |
| | Phi-3.5-mini-instruct | 53.25 | 60.04 | -6.79 | -4.75 | 0.000 | negative |
| | Llama-3.1-8B-Instruct | 57.87 | 37.79 | 20.08 | 13.92 | 0.000 | positive |
| | GPT-4o-mini | 53.92 | 43.67 | 10.25 | 7.10 | 0.000 | positive |
| | GPT-4o | 60.42 | 53.00 | 7.42 | 5.19 | 0.000 | positive |
| Arc_chall | Llama-3.2-3B-Instruct | 73.89 | 69.88 | 4.01 | 2.16 | 0.031 | positive |
| | Mistral-7B-Instruct-v0.3 | 76.45 | 73.63 | 2.82 | 1.58 | 0.115 | none |
| | Phi-3.5-mini-instruct | 86.52 | 83.36 | 3.16 | 2.14 | 0.032 | positive |
| | Llama-3.1-8B-Instruct | 85.32 | 79.52 | 5.80 | 3.69 | 0.000 | positive |
| | GPT-4o-mini | 93.77 | 90.61 | 3.16 | 2.85 | 0.004 | positive |
| | GPT-4o | 94.45 | 93.77 | 0.68 | 0.70 | 0.484 | none |
| Arc_easy | Llama-3.2-3B-Instruct | 84.39 | 83.46 | 0.93 | 0.87 | 0.383 | none |
| | Mistral-7B-Instruct-v0.3 | 86.99 | 84.68 | 2.31 | 2.28 | 0.022 | positive |
| | Phi-3.5-mini-instruct | 91.71 | 90.78 | 0.93 | 1.13 | 0.257 | none |
| | Llama-3.1-8B-Instruct | 89.73 | 88.64 | 1.09 | 1.21 | 0.226 | none |
| | GPT-4o-mini | 94.07 | 94.07 | 0.00 | 0.0 | 1.0 | none |
| | GPT-4o | 94.53 | 94.49 | 0.04 | 0.06 | 0.952 | none |

*Continued on next page*

**Token Signature: Predicting Chain-of-Thought Gains with Token Decoding Feature in Large Language Models**

|  | Model | CoT Acc | DA Acc | CoT Gain | Z Statistic | p Value | Significance |
|---|---|---|---|---|---|---|---|
| GPQA | Llama-3.2-3B-Instruct | 27.01 | 37.72 | -10.71 | -3.43 | 0.001 | negative |
|  | Mistral-7B-Instruct-v0.3 | 30.58 | 34.60 | -4.02 | -1.28 | 0.199 | none |
|  | Phi-3.5-mini-instruct | 23.21 | 16.29 | 6.92 | 2.60 | 0.009 | positive |
|  | Llama-3.1-8B-Instruct | 28.79 | 32.14 | -3.35 | -1.09 | 0.276 | none |
|  | GPT-4o-mini | 46.88 | 42.19 | 4.69 | -1.41 | 0.157 | none |
|  | GPT-4o | 61.16 | 49.33 | 11.83 | 3.56 | 0.000 | positive |
| MuSR | Llama-3.2-3B-Instruct | 44.84 | 50.93 | -6.09 | -2.37 | 0.018 | negative |
|  | Mistral-7B-Instruct-v0.3 | 45.90 | 47.35 | -1.45 | -0.57 | 0.572 | none |
|  | Phi-3.5-mini-instruct | 54.37 | 55.69 | -1.32 | -0.52 | 0.606 | none |
|  | Llama-3.1-8B-Instruct | 50.13 | 51.72 | -1.59 | -0.62 | 0.536 | none |
|  | GPT-4o-mini | 58.60 | 55.82 | 2.78 | 1.09 | 0.275 | none |
|  | GPT-4o | 62.96 | 59.39 | 3.57 | 1.42 | 0.154 | none |
| LSAT | Llama-3.2-3B-Instruct | 39.74 | 40.73 | -0.99 | -0.45 | 0.650 | none |
|  | Mistral-7B-Instruct-v0.3 | 45.00 | 46.58 | -1.58 | -0.71 | 0.473 | none |
|  | Phi-3.5-mini-instruct | 47.97 | 48.07 | -0.10 | -0.045 | 0.964 | none |
|  | Llama-3.1-8B-Instruct | 47.67 | 51.04 | -3.37 | -1.51 | 0.130 | none |
|  | GPT-4o-mini | 61.94 | 64.02 | -2.08 | -0.97 | 0.333 | none |
|  | GPT-4o | 75.92 | 72.65 | 3.27 | 1.68 | 0.093 | none |
| CSQA | Llama-3.2-3B-Instruct | 67.24 | 65.60 | 1.64 | 0.86 | 0.391 | none |
|  | Mistral-7B-Instruct-v0.3 | 70.93 | 69.12 | 1.81 | 0.98 | 0.329 | none |
|  | Phi-3.5-mini-instruct | 71.66 | 73.38 | -1.72 | -0.95 | 0.341 | none |
|  | Llama-3.1-8B-Instruct | 74.37 | 73.30 | 1.07 | 0.60 | 0.548 | none |
|  | GPT-4o-mini | 81.41 | 81.90 | -0.49 | -0.31 | 0.754 | none |
|  | GPT-4o | 84.11 | 84.68 | -0.57 | -0.39 | 0.698 | none |
| PIQA | Llama-3.2-3B-Instruct | 77.42 | 75.68 | 1.74 | 1.24 | 0.213 | none |
|  | Mistral-7B-Instruct-v0.3 | 80.20 | 81.77 | -1.57 | -1.21 | 0.225 | none |
|  | Phi-3.5-mini-instruct | 81.99 | 83.41 | -1.42 | -1.14 | 0.255 | none |
|  | Llama-3.1-8B-Instruct | 84.11 | 82.05 | 2.06 | 1.67 | 0.096 | none |
|  | GPT-4o-mini | 88.85 | 89.77 | -0.92 | -0.90 | 0.367 | none |
|  | GPT-4o | 91.08 | 93.63 | -2.55 | -2.91 | 0.004 | negative |
| SIQA | Llama-3.2-3B-Instruct | 62.69 | 62.33 | 0.36 | 0.23 | 0.816 | none |
|  | Mistral-7B-Instruct-v0.3 | 62.64 | 63.56 | -0.92 | -0.60 | 0.551 | none |
|  | Phi-3.5-mini-instruct | 70.57 | 70.01 | 0.56 | 0.38 | 0.702 | none |
|  | Llama-3.1-8B-Instruct | 69.50 | 68.68 | 0.82 | 0.55 | 0.579 | none |
|  | GPT-4o-mini | 77.23 | 76.00 | 1.23 | 0.91 | 0.364 | none |
|  | GPT-4o | 77.74 | 78.10 | -0.36 | -0.27 | 0.786 | none |
| StrategyQA | Llama-3.2-3B-Instruct | 66.59 | 79.69 | -13.1 | -10.00 | 0.000 | negative |
|  | Mistral-7B-Instruct-v0.3 | 86.16 | 82.93 | 3.23 | 3.02 | 0.002 | positive |
|  | Phi-3.5-mini-instruct | 75.46 | 73.89 | 1.57 | 1.22 | 0.222 | none |
|  | Llama-3.1-8B-Instruct | 71.88 | 76.55 | -4.67 | -3.61 | 0.000 | negative |
|  | GPT-4o-mini | 62.88 | 54.19 | 8.69 | 5.97 | 0.000 | positive |
|  | GPT-4o | 60.66 | 55.24 | 5.42 | 3.72 | 0.000 | positive |

Table 8: Instance SC, Aggregated SC, CoT Gain and CoT Gain Significance across benchmarks on Mistral-7B-Instruct-v0.3, Phi-3.5-mini-instruct, Llama-3.1-8B-Instruct.

**Mistral-7B-Instruct-v0.3**

|  | GSM8K | MultiArith | FOLIO | CH_a | CH_d | Arc_chall | Arc_easy |
|---|---|---|---|---|---|---|---|
| Instance SC | 0.1193 | 0.0837 | 0.1925 | -0.1410 | -0.035 | -0.1479 | -0.1218 |
| Aggregated SC | 0.4285 | 0.2781 | -0.0889 | -0.4540 | -0.509 | -0.6940 | -0.6640 |
| CoT Gain | 41.24 | 63.34 | -1.58 | 2.20 | -3.12 | 2.82 | 2.31 |
| Significance | positive | positive | none | none | negative | none | positive |

|  | GPQA | MuSR | LSAT | CSQA | PIQA | SIQA | StrategyQA |
|---|---|---|---|---|---|---|---|
| Instance SC | 0.0298 | 0.3173 | 0.1771 | -0.0488 | -0.0690 | 0.1708 | -0.1409 |
| Aggregated SC | -0.4995 | -0.4520 | -0.5528 | -0.4241 | -0.6690 | -0.7730 | -0.7609 |
| CoT Gain | -4.02 | -1.45 | -1.58 | 1.81 | -1.57 | -0.92 | 3.23 |
| Significance | none | none | none | none | none | none | positive |

**Phi-3.5-mini-instruct**

|  | GSM8K | MultiArith | FOLIO | CH_a | CH_d | Arc_chall | Arc_easy |
|---|---|---|---|---|---|---|---|
| Instance SC | 0.3292 | 0.2699 | 0.0437 | 0.0230 | 0.0098 | -0.1908 | -0.2007 |
| Aggregated SC | 0.6560 | 0.6160 | 0.2386 | 0.1460 | -0.0385 | -0.7586 | -0.7571 |
| CoT Gain | 59.89 | 40.84 | 5.82 | 10.71 | -6.79 | 3.16 | 0.93 |
| Significance | positive | positive | positive | positive | negative | positive | none |

|  | GPQA | MuSR | LSAT | CSQA | PIQA | SIQA | StrategyQA |
|---|---|---|---|---|---|---|---|
| Instance SC | 0.0015 | -0.0534 | 0.0126 | -0.2399 | -0.0839 | -0.0734 | -0.1905 |
| Aggregated SC | -0.3629 | -0.4933 | -0.4341 | -0.8655 | -0.6396 | -0.5340 | -0.7711 |
| CoT Gain | 6.92 | -1.32 | -0.10 | -1.72 | -1.42 | 0.56 | 1.57 |
| Significance | positive | none | none | none | none | none | none |

**Llama-3.1-8B-Instruct**

|  | GSM8K | MultiArith | FOLIO | CH_a | CH_d | Arc_chall | Arc_easy |
|---|---|---|---|---|---|---|---|
| Instance SC | 0.0015 | 0.1144 | 0.3419 | 0.236 | 0.094 | 0.2619 | 0.4176 |
| Aggregated SC | 0.2469 | 0.3061 | 0.5058 | 0.107 | 0.042 | -0.5972 | -0.5449 |
| CoT Gain | 66.95 | 49.50 | 8.80 | 9.71 | 20.08 | 5.80 | 1.09 |
| Significance | positive | positive | positive | positive | positive | positive | none |

|  | GPQA | MuSR | LSAT | CSQA | PIQA | SIQA | StrategyQA |
|---|---|---|---|---|---|---|---|
| Instance SC | 0.3395 | -0.0182 | 0.1033 | -0.0156 | 0.4174 | -0.0351 | -0.1669 |
| Aggregated SC | -0.3422 | -0.3472 | -0.2613 | -0.4229 | -0.1907 | -0.6066 | -0.1714 |
| CoT Gain | -3.35 | -1.59 | -3.37 | 1.07 | 2.06 | 0.82 | -4.67 |
| Significance | none | none | none | none | none | none | negative |

Table 9: Accuracy and token consumption of CoT, DA and Dynamic CoT experiments on Llama-3.2-3B-Instruct and Mistral-7B-Instruct-v0.3

| | CoT Acc | DA Acc | Dynamic CoT | CoT Tokens | DA Tokens | Dynamic CoT Tokens |
|---|---|---|---|---|---|---|
| **Llama-3.2-3B-Instruct** | | | | | | |
| GSM8K | **72.10** | 8.34 | 71.24 | 217.82 | 5.09 | 218.11 |
| MultiArith | **88.50** | 20.67 | 87.64 | 114.38 | 4.59 | 114.90 |
| FOLIO | **48.92** | 42.61 | 48.27 | 330.34 | 3.88 | 329.58 |
| CH_a | **35.21** | 33.54 | 32.26 | 250.42 | 3.53 | 45.862 |
| CH_d | **39.87** | 19.92 | 39.02 | 256.69 | 3.89 | 256.17 |
| Arc_chall | 73.89 | 69.88 | **74.51** | 238.37 | 4.03 | 238.07 |
| Arc_easy | **84.39** | 83.46 | 84.31 | 231.37 | 4.07 | 84.76 |
| GPQA | 27.01 | **37.72** | 34.42 | 729.99 | 4.03 | 4.03 |
| MuSR | 44.84 | **50.93** | 50.14 | 170.98 | 4.06 | 4.04 |
| LSAT | 39.74 | 40.73 | **40.77** | 386.84 | 4.01 | 128.73 |
| CSQA | **67.24** | 65.60 | 66.27 | 186.32 | 5.32 | 73.43 |
| PIQA | 77.42 | 75.68 | **77.68** | 178.08 | 4.94 | 178.47 |
| SIQA | 62.69 | 62.33 | **62.76** | 166.38 | 4.64 | 165.96 |
| StrategyQA | 66.59 | 79.69 | **79.87** | 212.67 | 4.28 | 4.28 |
| **Mistral-7B-Instruct-v0.3** | | | | | | |
| GSM8K | **49.58** | 8.34 | 47.83 | 241.32 | 8.60 | 242.04 |
| MultiArith | **77.67** | 14.33 | 76.00 | 157.25 | 7.49 | 158.24 |
| FOLIO | 49.42 | **51.00** | 50.69 | 113.75 | 7.75 | 7.77 |
| CH_a | **32.37** | 30.17 | 29.66 | 129.33 | 11.56 | 11.60 |
| CH_d | 55.92 | **59.04** | 55.87 | 125.95 | 9.95 | 125.77 |
| Arc_chall | 76.45 | 73.63 | **77.01** | 144.55 | 11.52 | 143.40 |
| Arc_easy | 86.99 | 84.68 | **87.49** | 119.36 | 9.41 | 118.47 |
| GPQA | 30.58 | **34.60** | 30.90 | 477.52 | 18.54 | 17.82 |
| MuSR | 45.90 | **47.35** | 46.32 | 220.77 | 9.69 | 65.74 |
| LSAT | 45.00 | **46.58** | 45.46 | 238.42 | 16.22 | 108.43 |
| CSQA | 70.93 | 69.12 | **71.31** | 112.47 | 5.67 | 110.77 |
| PIQA | 80.20 | 81.77 | **82.33** | 108.49 | 15.20 | 15.40 |
| SIQA | 62.64 | 63.56 | **63.97** | 111.61 | 8.28 | 31.24 |
| StrategyQA | 86.16 | 82.93 | **86.83** | 98.02 | 6.07 | 97.67 |

Table 10: Accuracy and token consumption of CoT, DA and Dynamic CoT experiments on Phi-3.5-mini-instruct and Llama-3.1-8B-Instruct

| | CoT Acc | DA Acc | Dynamic CoT | CoT Tokens | DA Tokens | Dynamic CoT Tokens |
|---|---|---|---|---|---|---|
| **Phi-3.5-mini-instruct** | | | | | | |
| GSM8K | **78.24** | 18.35 | 77.54 | 444.11 | 13.78 | 445.44 |
| MultiArith | 85.17 | 44.33 | **85.27** | 217.34 | 12.89 | 213.23 |
| FOLIO | 59.14 | 53.32 | **59.19** | 282.26 | 4.03 | 282.03 |
| CH_a | **40.29** | 33.58 | 39.66 | 262.88 | 4.74 | 262.86 |
| CH_d | 53.25 | 60.04 | **60.17** | 258.51 | 6.16 | 14.58 |
| Arc_chall | 86.52 | 83.36 | **87.25** | 329.93 | 4.69 | 330.22 |
| Arc_easy | 91.71 | 90.78 | **92.30** | 320.30 | 4.68 | 320.26 |
| GPQA | **23.21** | 16.29 | 19.10 | 661.04 | 5.86 | 670.99 |
| MuSR | 53.37 | 55.69 | **56.09** | 543.52 | 4.18 | 214.08 |
| LSAT | 47.97 | **48.07** | 47.24 | 426.53 | 5.17 | 425.43 |
| CSQA | 71.66 | 73.38 | **73.70** | 306.96 | 4.32 | 184.70 |
| PIQA | 81.99 | 83.41 | **84.17** | 308.52 | 4.08 | 4.08 |
| SIQA | **70.57** | 70.01 | 70.01 | 308.74 | 4.97 | 308.19 |
| StrategyQA | **75.46** | 73.89 | 74.64 | 272.98 | 4.05 | 51.60 |
| **Llama-3.1-8B-Instruct** | | | | | | |
| GSM8K | **81.20** | 14.25 | 80.54 | 218.91 | 11.51 | 218.14 |
| MultiArith | 91.00 | 41.50 | **91.09** | 118.16 | 7.01 | 117.93 |
| FOLIO | **58.22** | 49.42 | 58.06 | 323.22 | 4.56 | 322.66 |
| CH_a | **40.21** | 30.50 | 39.57 | 291.01 | 4.00 | 290.79 |
| CH_d | **57.87** | 37.79 | 57.19 | 242.38 | 4.00 | 241.49 |
| Arc_chall | 85.32 | 79.52 | **86.01** | 265.21 | 4.03 | 265.32 |
| Arc_easy | 89.73 | 88.64 | **90.50** | 230.45 | 4.07 | 194.21 |
| GPQA | 28.79 | **32.14** | 29.65 | 767.19 | 7.78 | 7.80 |
| MuSR | 50.13 | **51.72** | 51.42 | 299.08 | 4.00 | 4.00 |
| LSAT | 47.67 | **51.04** | 50.47 | 353.13 | 3.98 | 34.14 |
| CSQA | 74.37 | 73.30 | **74.89** | 224.82 | 4.04 | 224.50 |
| PIQA | 84.11 | 82.05 | **85.07** | 168.42 | 4.02 | 144.47 |
| SIQA | 69.50 | 68.68 | **69.80** | 198.46 | 4.01 | 185.58 |
| StrategyQA | 71.88 | 76.55 | **76.83** | 189.51 | 4.31 | 4.28 |

Table 11: Accuracy and token consumption of CoT, DA and Dynamic CoT(transfer) experiments on closed-source models

| | | | GPT-4o-mini | | | |
|---|---|---|---|---|---|---|
| | CoT Acc | DA Acc | Dynamic CoT | CoT Tokens | DA Tokens | Dynamic CoT Tokens |
| GSM8K | **84.38** | 31.01 | **84.38** | 314.94 | 5.71 | 314.94 |
| MultiArith | **94.33** | 94.83 | **94.33** | 193.42 | 5.14 | 193.42 |
| FOLIO | **69.19** | 62.21 | **69.19** | 360.00 | 3.00 | 359.30 |
| CH_a | **56.92** | 46.25 | **56.92** | 360.33 | 3.01 | 360.16 |
| CH_d | **53.92** | 43.67 | **53.92** | 296.18 | 3.01 | 296.18 |
| Arc_chall | **93.77** | 90.61 | **93.77** | 255.05 | 3.06 | 255.05 |
| Arc_easy | **94.07** | 94.07 | **94.07** | 231.77 | 3.06 | 231.61 |
| GPQA | **46.88** | 42.19 | 46.43 | 667.46 | 3.58 | 9.75 |
| MuSR | **58.60** | 55.82 | 55.82 | 406.48 | 2.98 | 4.16 |
| LSAT | 61.94 | **64.02** | 63.33 | 488.74 | 3.01 | 234.38 |
| CSQA | 81.41 | **81.90** | 81.41 | 212.58 | 3.00 | 212.19 |
| PIQA | 88.85 | **89.77** | 88.79 | 208.04 | 2.99 | 150.36 |
| SIQA | **77.23** | 76.00 | **77.23** | 233.87 | 3.02 | 232.90 |
| StrategyQA | **62.88** | 54.19 | 54.24 | 206.40 | 3.00 | 3.17 |

| | | | GPT-4o | | | |
|---|---|---|---|---|---|---|
| | CoT Acc | DA Acc | Dynamic CoT | CoT Tokens | DA Tokens | Dynamic CoT Tokens |
| GSM8K | **85.14** | 53.90 | **85.14** | 314.56 | 5.86 | 314.56 |
| MultiArith | 93.50 | **98.00** | 93.50 | 185.56 | 5.54 | 185.56 |
| FOLIO | **75.08** | 69.60 | **75.08** | 523.60 | 3.00 | 522.48 |
| CH_a | **59.42** | 42.13 | **59.42** | 599.96 | 3.00 | 599.70 |
| CH_d | **60.42** | 53.00 | **60.42** | 425.53 | 2.98 | 425.53 |
| Arc_chall | **94.45** | 93.77 | **94.45** | 372.28 | 3.03 | 372.28 |
| Arc_easy | **94.53** | 94.49 | **94.53** | 345.25 | 3.05 | 344.99 |
| GPQA | **61.16** | 49.33 | 49.33 | 883.89 | 3.77 | 10.78 |
| MuSR | **62.96** | 59.39 | 59.39 | 629.73 | 3.01 | 4.96 |
| LSAT | **75.92** | 72.65 | 73.84 | 690.96 | 3.04 | 341.13 |
| CSQA | 84.11 | **84.68** | 84.03 | 300.50 | 3.00 | 299.94 |
| PIQA | 91.08 | **93.63** | 91.78 | 273.49 | 2.96 | 197.62 |
| SIQA | 77.74 | **78.10** | 77.69 | 247.44 | 3.02 | 246.60 |
| StrategyQA | **60.66** | 55.24 | 55.28 | 267.89 | 2.99 | 3.19 |

Table 12: Prediction accuracy of two CoT effectiveness indicators (Instance SC and aggregated SC) using Llama-3.2-3B-Instruct with modified decoding strategies

| Strategies | temperature = 0.3 | | | temperature = 0.7 | | | temperature = 0.9 | | |
|---|---|---|---|---|---|---|---|---|---|
| | topk=5 | topk=10 | topk=20 | topk=5 | topk=10 | topk=20 | topk=5 | topk=10 | topk=20 |
| **Agg_accuracy(%)** | 92.86 | 92.86 | 85.71 | 85.71 | 85.71 | 85.71 | 85.71 | 85.71 | 85.71 |
| **Ins_accuracy(%)** | 85.71 | 85.71 | 78.57 | 92.86 | 92.86 | 92.86 | 92.86 | 92.86 | 92.86 |

Table 13: Experiments based on few-shot CoT: Instance SC, Aggregated SC, CoT Gain and CoT Gain Significance across benchmarks on Llama-3.2-3B-Instruct, Mistral-7B-Instruct-v0.3, Phi-3.5-mini-instruct, Llama-3.1-8B-Instruct.

| | GSM8K | MultiArith | FOLIO | CH_a | CH_d | Arc_chall | Arc_easy |
|---|---|---|---|---|---|---|---|
| **Llama-3.2-3B-Instruct** | | | | | | | |
| Instance SC | 0.0450 | 0.1619 | 0.1869 | 0.0720 | 0.0900 | -0.0513 | -0.0552 |
| Aggregated SC | 0.2080 | 0.2904 | 0.0488 | -0.032 | 0.2670 | -0.4597 | -0.6061 |
| CoT Gain | 67.25 | 72.16 | 6.31 | 7.54 | 26.7 | -1.36 | -3.28 |
| Significance | positive | positive | positive | positive | positive | none | negative |
| | GPQA | MuSR | LSAT | CSQA | PIQA | SIQA | StrategyQA |
| Instance SC | 0.0038 | -0.0840 | -0.0661 | -0.1102 | -0.2698 | -0.2698 | -0.0082 |
| Aggregated SC | -0.1042 | -0.4016 | -0.4180 | -0.3424 | -0.5340 | -0.7639 | -0.3917 |
| CoT Gain | -7.59 | -5.69 | -4.46 | 0.82 | 0.44 | 2.77 | -6.46 |
| Significance | negative | negative | negative | none | none | none | negative |
| **Mistral-7B-Instruct-v0.3** | | | | | | | |
| | GSM8K | MultiArith | FOLIO | CH_a | CH_d | Arc_chall | Arc_easy |
| Instance SC | 0.1193 | 0.0837 | 0.1925 | -0.1410 | -0.035 | -0.1479 | -0.1218 |
| Aggregated SC | 0.4285 | 0.2781 | -0.0889 | -0.4540 | -0.509 | -0.6940 | -0.6640 |
| CoT Gain | 40.86 | 59.34 | -2.58 | 2.83 | -1.79 | 2.22 | 0.34 |
| Significance | positive | positive | none | positive | none | none | none |
| | GPQA | MuSR | LSAT | CSQA | PIQA | SIQA | StrategyQA |
| Instance SC | 0.0298 | 0.3173 | 0.1771 | -0.0488 | -0.0690 | 0.1708 | -0.1409 |
| Aggregated SC | -0.4995 | -0.4520 | -0.5528 | -0.4241 | -0.6690 | -0.7730 | -0.7609 |
| CoT Gain | -5.36 | -0.79 | -0.99 | -0.73 | -3.04 | -2.10 | 5.28 |
| Significance | none | none | none | none | negative | none | positive |
| **Phi-3.5-mini-instruct** | | | | | | | |
| | GSM8K | MultiArith | FOLIO | CH_a | CH_d | Arc_chall | Arc_easy |
| Instance SC | 0.3292 | 0.2699 | 0.0437 | 0.0230 | 0.0098 | -0.1908 | -0.2007 |
| Aggregated SC | 0.6560 | 0.6160 | 0.2386 | 0.1460 | -0.0385 | -0.7586 | -0.7571 |
| CoT Gain | 59.36 | 43.50 | 0.25 | 1.63 | -0.33 | -1.88 | 0.55 |
| Significance | positive | positive | none | none | none | none | none |
| | GPQA | MuSR | LSAT | CSQA | PIQA | SIQA | StrategyQA |
| Instance SC | 0.0015 | -0.0534 | 0.0126 | -0.2399 | -0.0839 | -0.0734 | -0.1905 |
| Aggregated SC | -0.3629 | -0.4933 | -0.4341 | -0.8655 | -0.6396 | -0.5340 | -0.7711 |
| CoT Gain | 4.69 | 1.06 | 0.49 | 0.66 | -3.37 | -1.07 | 4.58 |
| Significance | none | none | none | none | negative | none | positive |
| **Llama-3.1-8B-Instruct** | | | | | | | |
| | GSM8K | MultiArith | FOLIO | CH_a | CH_d | Arc_chall | Arc_easy |
| Instance SC | 0.0015 | 0.1144 | 0.3419 | 0.236 | 0.094 | 0.2619 | 0.4176 |
| Aggregated SC | 0.2469 | 0.3061 | 0.5058 | 0.107 | 0.042 | -0.5972 | -0.5449 |
| CoT Gain | 66.11 | 55.17 | 9.13 | 13.54 | 22.46 | 4.01 | 0.16 |
| Significance | positive | positive | positive | positive | positive | positive | none |
| | GPQA | MuSR | LSAT | CSQA | PIQA | SIQA | StrategyQA |
| Instance SC | 0.3395 | -0.0182 | 0.1033 | -0.0156 | 0.4174 | -0.0351 | -0.1669 |
| Aggregated SC | -0.3422 | -0.3472 | -0.2613 | -0.4229 | -0.1907 | -0.6066 | -0.1714 |
| CoT Gain | 3.13 | -2.12 | 0.40 | 0.25 | 0.27 | -0.41 | -2.18 |
| Significance | none | none | none | none | none | none | none |

