# OpenReview forum: "Token Signature: Predicting Chain-of-Thought Gains with Token Decoding Feature in Large Language Models"
_ICML.cc/2025/Conference — ICML 2025 poster_

### Official Review · Reviewer_xvNF · 2025-03-07

**Overall Recommendation:** 3

**Summary:**

The author's overall approach is simple and clear: CoT is not always necessary → Token Signature is correlated with CoT gain → The token probability distribution can be used to determine whether CoT is needed. Through experiments, the study validates that Aggregated SC can effectively indicate whether a task benefits from CoT, and the experimental results are quite promising.

**Claims And Evidence:**

Claim: Token Signature can help classify tasks and dynamically decide whether to use CoT reasoning, thereby maintaining high accuracy while reducing unnecessary token consumption.

-> This method was tested on multiple tasks across different benchmark datasets, demonstrating a strong correlation between Token Signature and CoT gain.

**Essential References Not Discussed:**

No

**Experimental Designs Or Analyses:**

The experimental design is solid, testing Token Signature and Dynamic CoT across multiple benchmarks and models. The correlation between token probability trends and CoT effectiveness is well-supported, but causality isn’t fully established. It’d help to see ablation studies and failure case analysis to understand when Dynamic CoT misclassifies. Also, testing more closed-source models beyond GPT-4o would strengthen generalization claims.

**Methods And Evaluation Criteria:**

The proposed methods and evaluation criteria seem pretty reasonable for the problem they’re tackling. The idea of using Token Signature to predict whether CoT will be effective makes sense, and they back it up with solid benchmarks across different types of tasks. The Aggregated SC metric is a smart way to analyze token probability trends, and the Dynamic CoT method helps optimize when to use CoT, balancing accuracy and efficiency.

The evaluation setup is well thought out—they test on a variety of benchmarks (math, commonsense, symbolic reasoning, etc.) and compare against both open-source and closed-source models. That gives a broad perspective on how well the method generalizes. One minor thing is that they only tested closed-source transfer on GPT-4o, so it’d be nice to see results on more proprietary models. But overall, the approach makes sense, and the experiments support their claims well.

**Other Comments Or Suggestions:**

Figure 1 - CoT is significantly greater than 0 (p ¡ 0.05).
Predicting Chain-of-Thought Gains with Token Decoding Feature" - Should "Feature" be plural (Features)?

**Other Strengths And Weaknesses:**

The paper is easy to follow.

**Questions For Authors:**

While the experiments show good results, is there a theoretical basis for why Token Signature correlates with CoT gain? Could information theory or Bayesian updating explain this relationship? Have you considered other indicators (e.g., token confidence variance, entropy trends) to compare against Aggregated SC? Are there cases where Token Signature mispredicts CoT effectiveness? A breakdown of such cases could strengthen the analysis.

**Relation To Broader Scientific Literature:**

Chain-of-Thought is a widely used trick for LLMs, so this paper should be quite useful for improving overall LLM applications.

**Theoretical Claims:**

No theoretical claims

---

> ### Author Rebuttal · Authors · 2025-04-01
>
> ## Response to reviewer xvNF：
> Dear reviewer:
> Thank you for reviewing our article and raising questions and suggestions. We now respond to the corresponding questions as follows.
>
> ### Q1: They only tested closed-source transfer on GPT-4o, so it’d be nice to see results on more proprietary models.
> **Response:** Thanks for your suggestion. Testing more proprietary models will increase the robustness of our approach. We will include testing of other proprietary models in the latest version of the paper.
>
> ### Q2: It’d help to see ablation studies and failure case analysis to understand when Dynamic CoT misclassifies.
> **Response:** Thanks for your suggestion. We perform some ablation experiments to strengthen our research, such as the prediction accuracy of the SC metric of the first N tokens decoded, and changing the prompt (few-shot CoT/zero-shot CoT) to see the impact. In order to verify the impact of different parameters, we use 10/20/50/100/200 tokens for experiments, and compare the prediction performance of Instance SC and Aggregated SC respectively. The comparison shows that the prediction performance of the first 50 tokens is the best (Instance SC (69.6%) and Aggregated_SC (89.3%)). We also explore the impact of different types of CoT, and the results show the robustness of our method. The above ablation experiment will be updated in the latest version of our paper. In addition, we will also add failure case analysis to the latest version of the paper to analyze the possible causes of failure (classification training samples/random errors, etc.) to ensure the comprehensiveness of our research.
>
> ### Q3: Other writing problems in Figure 1.
> **Response:** Thank you for pointing out the writing problems in the paper. We re-checked the paper and corrected the pointed out problems.
>
> ### Q4: Is there a theoretical basis for why Token Signature correlates with CoT gain?
> **Response:** Thanks for your question. We give the following theoretical analysis:
> Token probability can reflect LLM’s confidence in an answer [1]. Chain-of-thought allows for improving confidence for inherent serial problems by spending more tokens for deep search [2]. However, other not inherently sequential problems might have negative outcomes because of snowball errors [3]. In other words, the model's low confidence may steer it down the wrong reasoning path. When CoT is introduced, the snowball effect is exacerbated, amplifying the initial error further. By incorporating the Spearman correlation indicator, we can assess the model's uncertainty at the start of the reasoning process, helping us determine whether applying CoT is beneficial. Therefore, SC is a good metric to predict the performance gain of CoT. We will make the above content clearer in the latest version of the paper.
> [1] Detecting hallucinations in large language models using semantic entropy. Nature 2024
> [2] Chain of thought empowers transformers to solve inherently serial problems. arXiv 2024
> [3] Rethinking External Slow-Thinking: From Snowball Errors to Probability of Correct Reasoning. arXiv 2025
>
> ### Q5: Have you considered other indicators (e.g., token confidence variance, entropy trends) to compare against Aggregated SC?
> **Response:** Thanks for your question. We explore multiple indicators in our initial exploration, such as the Mean of Absolute Increments and Probabilistic Entropy. For individual cases, the probability distribution of tokens often has a certain degree of divergence. Based on our large number of experimental observations, we believe that these two indicators are difficult to measure the characteristics of different questions when the model answers. For the CoT gain judgment at the benchmark granularity level, we find that the Mean of Absolute Increments is often affected by changes in the number of questions, and it is difficult for us to avoid errors caused by randomness, so we discard it. We observe that the average token probability distribution trend of the initial token is not the same (Figure 2 in the paper), and it is also highly correlated with the question category. There is no work to study it from the perspective of the overall decoding distribution. Inspired by this, we use Spearman Correlation as the main indicator. I hope our answer can address your concerns. And we will add relevant discussions on other exploration indicators in the latest version of the paper.

---

### Official Review · Reviewer_YmMf · 2025-03-12

**Overall Recommendation:** 4

**Summary:**

This paper presents token signature, i.e., the spearman correlation between token probability distributions and token indices, that can better help decide if a CoT is needed for a specific task or not. The authors further introduced dynamic CoT, that can do online selection of whether to use CoT or direct answer, and showed accuracy and efficiency improvements over multiple tasks.

**Claims And Evidence:**

Yes, most claims are well supported.

- For the spearman correlation, the authors presented a few examples, is there any intuitive explanation on why this might work? Also, how much is this correlation affected by the CoT prompt (e.g., if I use few-shot CoT instead of zero-shot CoT)?

**Essential References Not Discussed:**

No.

**Experimental Designs Or Analyses:**

Yes the experiments are fairly comprehensive.

- Evaluation is done over 4 model families and 5 reasoning categories.
- The correlation between instance/aggregated SC and CoT gain is well illustrated.
- In addition to accuracy improvements, the authors also showed token consumption, indicating efficiency improvements using dynamic CoT.

**Methods And Evaluation Criteria:**

The methods are reasonable and evaluation is done over multiple language models (Llama, Phi, Mistral, GPT-4o) and over a wide set of datasets (math, symbolic, knowledge, commonsense etc).

**Other Comments Or Suggestions:**

See above.

**Other Strengths And Weaknesses:**

- Overall the idea is novel. If possible can the authors add more intuition on why the spearman correlation between token probability and token index can help decide when a CoT is needed or not, other than empirical observations?

- The experiments overall are sound. From Table 3 and 4, it's clear that the Dynamic CoT can achieve a very good balance between performance from using a CoT and not using a CoT, and the results transfer well to closed-source models. Figure 4 and 5 present token efficiency improvements, which is another benefit of adopting dynamic CoT.

**Questions For Authors:**

See above. Specifically, can the authors add more intuition on why the SC is correlated with whether CoT is needed or not?

**Relation To Broader Scientific Literature:**

Overall the idea is quite novel. The authors observed an interesting phenomenon that the token probability distribution has a positive correlation with token index when CoT is used (and vice versa for direct answers), and proposed a reasonable method to dynamically select whether to use CoT or not for a wide set of tasks.

**Theoretical Claims:**

N/A

---

> ### Author Rebuttal · Authors · 2025-04-01
>
> ## Response to reviewer YmMf：
> Dear reviewer:
> Thank you for reviewing our article and raising questions and suggestions. We now respond to the corresponding questions as follows.
>
> ### Q1: For the spearman correlation, the authors presented a few examples, is there any intuitive explanation on why this might work?
> **Response:** Spearman correlation (SC) measures the monotonic relationship between the initial token probability distribution and the sequence order during decoding. It works intuitively for the following reasons:
> Token probability can reflect LLM’s confidence in an answer [1]. Chain-of-thought allows for improving confidence for inherent serial problems by spending more tokens for deep search [2]. However, other not inherently sequential problems might have negative outcomes because of snowball errors [3]. Because math tasks (such as GSM8K) have a strict step-dependent structure, such as arithmetic operations need to be performed in sequence, and logical deductions need to be carried out step by step. The solution space of such tasks is highly constrained, and the generation of intermediate steps needs to follow clear rules. By introducing CoT, a method similar to deep search, the certainty of the model's answer can be well enhanced. For other problems, such as common sense reasoning, the solution space is relatively diverse. When the initial confidence of the model is not high, it is easier for the model to fall into the wrong reasoning path. The introduction of CoT will gradually amplify the error due to the snowball effect. By incorporating the Spearman correlation indicator, we can assess the model's uncertainty at the start of the reasoning process, helping us determine whether applying CoT is beneficial. Therefore, SC is a good metric to predict the performance gain of CoT.
>
> [1] Detecting hallucinations in large language models using semantic entropy. Nature 2024
> [2] Chain of thought empowers transformers to solve inherently serial problems. arXiv 2024
> [3] Rethinking External Slow-Thinking: From Snowball Errors to Probability of Correct Reasoning. arXiv 2025
>
> ### Q2: How much is the correlation affected by the CoT prompt (use few-shot CoT instead of zero-shot CoT)?
> **Response:** Thanks for your concern. We have included the impact of other prompt settings, such as few-shot CoT. The results show that for few-shot CoT, our method also achieves high prediction accuracy. We use standard prompts (i.e., only provide required questions as prompts) to calculate the proposed Spearman correlation metric, and then predict the CoT gain at the benchmark granularity. We focus on zero-shot CoT as the main research object and supplement the experiment with few-shot CoT. The experimental results show that the gains of few-shot CoT and zero-shot CoT are highly consistent, so at the benchmark granularity, the proposed method can also well predict the gain of few-shot CoT, and the experimental results are very robust. We hope our answer can address your concern. We will add detailed experimental results as supplementary material in the latest version of the paper.

---

### Official Review · Reviewer_revN · 2025-03-14

**Overall Recommendation:** 2

**Summary:**

The paper makes  this observation that in certain tasks (where CoT) is known to help, the probability of the token predicted generally increases as more and more tokens are generated. They propose to exploit this observation to predict whether CoT would help on a task or not.

**Claims And Evidence:**

They give some evidence into why their hypothesis might be true. But it's limited to GSM8k level reasoning tasks.

**Essential References Not Discussed:**

NA

**Experimental Designs Or Analyses:**

Yes, I checked. See weakness section.

**Methods And Evaluation Criteria:**

See weakness section

**Other Comments Or Suggestions:**

See weakness section.

**Other Strengths And Weaknesses:**

Weakness:

- I fail to understand the proposed approach. First, even to predict whether CoT would help or not, they need to generate tokens to get the \(\rho_i\) variable. Thus, you anyways will end up spending compute on each and every test example.

- However, the key issue is that the whole work is based on observations on a few benchmarks where it is well known that CoT helps or not. For today’s frontier tasks (e.g., AIME, MATH) where models actually reason for long (e.g., for up to thousands of tokens), such kind of monotonic behaviors might not hold at all.

- More crucially, the paper doesn’t share absolutely any reasonable understanding or justification about why token probabilities might increase for math tasks and not for others. L282-287 is not justification, simply stating the algorithm in words.

- The writeup of section 5.2 seems quite misleading. The authors mention DirectCoT is ranked amongst top 2 methods in 92% cases, albeit there are in total 3 methods in question. Moreover, in most of the tasks they consider in this paper and Table 8, there is not a huge difference between CoT and non-CoT (called DA in the paper) performance.

- In Figure 4 and Figure 5, the authors try to mention that their approach consumes fewer tokens. However, many tasks in the Figure are in fact MCQ, where token consumption is naturally low, making the comparison misleading.

**Questions For Authors:**

See weakness section.

**Relation To Broader Scientific Literature:**

I believe the community at this point is focused on how to prevent overthinking of LLMs in many tasks. While the proposed approach in this paper might not scale to harder tasks with extremely long CoTs, it does related to that line of work.

**Theoretical Claims:**

NA

---

> ### Author Rebuttal · Authors · 2025-04-01
>
> ## Response to reviewer revN：
> Dear reviewer:
> Thank you for reviewing our article and raising questions and suggestions. We now respond to the corresponding questions as follows.
>
> ### Q1: Explanation of the proposed method.
> **Response:** Thanks for your question. We will explain it from the following two aspects:
> 1. The proposed method does not need to decode all tokens when LLM answers, but it adopts the method of decoding some tokens. For example, we use the probability of decoding the first N tokens to judge the CoT effectiveness of the task-category in the paper, which does not consume too many computing resources, and our proposed indicator achieves up to 89% CoT prediction accuracy at the task-category level.
> 2. The method we proposed can be transferred to other models. Through integration in some small models, it can be well transferred to other proprietary models without spending additional computing resources. Our results show that the transferred method can reduce token consumption by more than 35% while achieving high accuracy.
> We will make the above content clearer in the latest version of the paper.
>
> ### Q2: The benchmarks are limited to tasks where CoT is known to help, limiting generalizability to complex tasks like AIME or MATH.
> **Response:** We appreciate your suggestion to validate the method on other, more complex long-term reasoning tasks such as MATH. The current experiments choose classic benchmarks to ensure reproducibility and comparability. It should be emphasized that the core idea of Token Signature - dynamically selecting policies through confidence trends during decoding - is not limited to sequence length because we only decode the first N tokens. For example, even in long sequences, the initial token distribution (such as the first 50 tokens) may still reflect the model's early confidence in step-by-step reasoning. For AIME, we believe that it is a benchmark that tests the reasoning ability of inference models now, which may not be suitable for our small model. The results on MATH are consistent with the experiments in this paper. Aggregated SC and Instance SC are greater than 0, and it is predicted that CoT has a positive gain on MATH. We will include relevant experiments on the MATH benchmark in the latest version of the paper.
>
> |            | Llama-3.2 | Phi-3.5 | Mistral-7B | Llama-3.1 |
> |------------|-----------|---------|------------|-----------|
> | Instance SC| 0.0789    | 0.0859  | 0.0473     | 0.0557    |
> | Aggregated SC| 0.2479   | 0.1834  | -0.1234    | 0.4829    |
>
> ### Q3: The paper doesn’t share a reasonable justification for why token probabilities might increase for math tasks and not for others.
> **Response:** Thanks for your question. Due to character limitations, we also answer similar issues in reviewers YmMf(Q1) and reviewers xvNF(Q4). We recommend that you check the comments of other reviewers.
>
> ### Q4: The write-up of section 5.2 seems quite misleading. In Table 8, there is no difference in the performance of CoT and DA for most tasks.
> **Response:** Thanks for your concern.
> “Dynamic CoT ranks in the top 2 in 92.8% of cases” is intended to show that it can match or surpass better policies in CoT and DA, rather than absolutely outperforming all possible methods. To reduce possible misleading, we have revised it to: “Dynamic CoT achieves comparable accuracy to better policies in CoT or DA in 92.8% of experimental settings.”
> For many tasks in Table 8, there is not much difference between the performance of CoT and DA, but CoT consumes more tokens than DA, which also makes a dynamic judgment of CoT particularly important for improving efficiency and reducing unnecessary expenses. Our method aims to maintain its answer accuracy while minimizing the use of tokens. Specifically, our method can effectively reduce token usage by more than 35%.
>
> ### Q5: Token consumption comparisons can be misleading.
> **Response:** Thanks for your concern. For our experiments, whether it is Short Answer or Multiple Choice, the token usage of the corresponding experiments (CoT & DA) remains at the same level. For the CoT experiment, the token usage is around a few hundred tokens. For the DA experiment, we designed the prompt to limit the model to only output the final answer without the intermediate process, and its token usage is within 10 tokens. Therefore, our comparison is relatively fair. In order to reduce possible doubts about this aspect, we will add the specific token usage of each group of experiments in the latest version of the paper to make the research more transparent.

---

> > ### Comment · Reviewer_revN · 2025-04-04
> >
> > I have read the author response. I still do not get the basic motivation, as it's not even hard to predict on what tasks CoT would help intuitively. Moreover, reasoning helps in ways beyond just accuracy as well these days. A lot of experiment settings in the paper are still not clear after the author response (ex. Q5). I maintain my rating.

---

> > > ### Author Response · Authors · 2025-04-09
> > >
> > > **Dear Reviewer,**
> > >
> > > Thank you again for your valuable feedback. We would like to take this opportunity to further address your concerns and explain the motivation of our work.
> > >
> > > ### **Motivation**
> > >
> > > - We aim to explore why Chain-of-Thought (CoT) reasoning provides inconsistent performance gains across tasks and identify predictive signals for determining when CoT is likely to be beneficial.
> > >
> > > - **We seek to reduce unnecessary CoT usage, particularly in scenarios where it provides little or no benefit, by dynamically deciding whether CoT or a Direct Answer (DA) is more suitable for each task.** This enables us to save computational resources without compromising model performance.
> > >
> > > ### **Method Summary**
> > >
> > > Our approach is designed to address the aforementioned goals in the following manner:
> > >
> > > - **Transferring from small language models to large language models:** We extend our approach to large language models using the token signature derived from small language models. This allows us to further significantly reduce the computation cost of large language models and maintain high prediction accuracy. Our experiments demonstrate that Dynamic CoT can reduce token consumption by over 33.3% without sacrificing performance.
> > >
> > > - **Token Signature as Early Predictor:** We observe the probability distribution trends of the first 50 tokens generated during decoding (typically covering 28% of the entire response). We discover that the shape of this distribution (monotonicity) is highly correlated with CoT effectiveness. **Therefore, the signature of the first few tokens can be the early predictor of the necessity of CoT, saving most of the computation of decoding each and every tokens .**
> > >
> > > ### **Response to Q5: Potential Misleading Comparison Due to MCQ Format**
> > >
> > > We understand your concern that the token consumption comparison in Figures 4 and 5 might be biased due to the nature of multiple-choice questions (MCQ). To address this concern and ensure a fair comparison:
> > >
> > > - We have restructured the GSM8K and MultiArith benchmarks into MCQ format. This standardization eliminates the potential bias introduced by different answer formats (e.g., short answers versus multiple-choice).
> > >
> > > - We have provided a more detailed breakdown of token consumption for each experiment (CoT, DA, and Dynamic CoT) across the four language models. The results show that Dynamic CoT can reduce token consumption by 35.9%. Detailed results can be found at [Token Consumption Comparison PDF](https://anonymous.4open.science/r/token_signature_rebuttal-59CB/Token%20consumption%20comparison.pdf).
> > >
> > > We hope that this additional clarification addresses your concern regarding the setting of our evaluation and helps to further validate the robustness of our findings.
> > >
> > >
> > > Once again, we greatly appreciate your time and the opportunity to improve our paper. We believe that these updates clarify both the motivations behind our approach and the fairness of our experimental design, and we look forward to hearing your thoughts on these revisions.
> > >
> > > Sincerely,
> > >
> > > The Authors

---

### Official Review · Reviewer_qdHF · 2025-03-15

**Overall Recommendation:** 2

**Summary:**

This paper examines the inconsistency of Chain-of-Thought (CoT) reasoning across different tasks and introduces Token Signature, a novel approach for predicting CoT effectiveness based on token probability distributions. The authors develop two key evaluation metrics, Instance Spearman Correlation (Instance SC) and Aggregated Spearman Correlation (Aggregated SC), to quantify CoT reliability. Additionally, they propose Dynamic CoT, a logistic regression-based method for adaptively selecting between CoT reasoning and direct answers. Extensive experiments on various benchmarks validate the effectiveness of these approaches.

## update after rebuttal
I'd like to thank the authors for their rebuttal. After reading the response, I still feel that this paper can be significantly strengthened by including thorough analysis and justification to their technical designs, which is nontrivial and missing from the current manuscript. Given that, I would like to keep my rating unchanged.

**Claims And Evidence:**

The claims made in this paper are generally well-supported by the experimental results. The proposed Instance SC and Aggregated SC metrics demonstrate strong predictive performance, achieving 69.7% and 87.5% accuracy, respectively, in assessing CoT effectiveness. Furthermore, the Dynamic CoT approach consistently matches or outperforms the best performance between CoT and direct answers while significantly reducing token consumption.

**Essential References Not Discussed:**

To the best of my knowledge, no essential references are missing from the discussion.

**Experimental Designs Or Analyses:**

The experimental design is thorough, incorporating evaluations across multiple LLMs, including both open-source and closed-source models, and a diverse set of benchmarks.

**Methods And Evaluation Criteria:**

The Token Signature approach is well-defined, utilizing Spearman correlation to assess CoT gains, and the evaluation is strengthened by benchmarks covering a diverse range of tasks, including mathematical, symbolic, commonsense, and reasoning challenges.

However, several critical technical design choices lack justification. For instance, in Lines 174–176, the authors restrict the indicator score computation to only the first 50 tokens, without explaining why this specific cutoff is appropriate. Additionally, in Lines 312–317, the decision to classify CoT effects based on the boundary values -2 and 2 appears arbitrary, and in Lines 317–319, the indicator is deemed accurate if the indicator value > 0 and CoT gain > 2, or if the indicator value < 0 when CoT < 2, introducing an asymmetric threshold without clear rationale. A justification for these choices, along with a sensitivity analysis, would be necessary to confirm their robustness and impact on the final results.

**Other Comments Or Suggestions:**

The writing quality needs significant improvement, particularly in terms of language use and formatting. Issues such as improper capitalization (e.g., Line 20: “In this work, We initially …”) and missing punctuation (e.g., Line 30: “high accuracy Overall, we …”) are prevalent throughout the paper.

**Other Strengths And Weaknesses:**

Another concern is that the paper provides limited discussion on decoding strategies, particularly regarding how temperature settings or sampling strategies might influence the effectiveness of Token Signature. Since token probability distributions can be highly sensitive to decoding parameters, an analysis of their impact would strengthen the evaluation and clarify whether the proposed method remains robust across different inference settings.

**Questions For Authors:**

The motivation presented in Lines 43–48 is unclear and somewhat contradictory. The authors state that “the effectiveness of CoT across different problems and models can be generally inferred from the task category,” yet they also argue that there is no definitive measure of effectiveness, necessitating their proposed methodology. If CoT effectiveness can already be inferred at the task-category level, what is the additional benefit the proposed task-level effectiveness measure provides?

**Relation To Broader Scientific Literature:**

The paper builds on prior work in CoT reasoning and LLM decoding strategies, but its novelty lies in the predictive framework for CoT effectiveness, which helps identify when and why CoT reasoning is beneficial.

**Theoretical Claims:**

The paper introduces the hypothesis that token probability distributions correlate with CoT gains, which is supported by empirical results. However, it lacks an in-depth theoretical analysis. A more rigorous formalization or theoretical justification of why token probability distributions should predict CoT effectiveness would strengthen the paper’s contributions and provide deeper insights into the underlying mechanisms driving the observed correlations.

---

> ### Author Rebuttal · Authors · 2025-04-01
>
> ## Response to reviewer qdHF：
> Dear reviewer:
> Thank you for reviewing our article and raising questions and suggestions. We now respond to the corresponding questions as follows.
>
> ### Q1: Several critical technical designs.
> 1) Restricting the indicator score computation to only the first 50 tokens in Lines 174–176.
> **Response:** We use the first 50 tokens, which is an empirical parameter. We believe that if this parameter is set too small, the correlation trend will not be obvious, and if the parameter is set too large, the data at high token index will be sparse due to the existence of short decoding paths. In general, taking 50 as our experimental setting is a comprehensive consideration. In order to verify the impact of different parameters, we use 10/20/50/100/200 tokens for experiments, and compare the prediction performance of Instance SC and Aggregated SC respectively. The comparison show that the prediction performance of the first 50 tokens is the best (Instance SC (69.6%) and Aggregated SC (89.3%)). We will add parameter experiments in the supplementary materials and clearly state the reason for setting the threshold in the main text. Thank you for your advice.
>
> 2) About CoT gain threshold in Lines 312–319.
> **Response:** Thanks for your question. We use 2 as the critical point to judge the CoT gain in order to solve the randomness problem in the experiment. In order to make the experiment more reasonable, we introduce the z-test and improve the evaluation method. We combine the number of questions N, CoT_Acc, and DA_Acc in the benchmark, and use a two-sided z-test to test the significance of CoT gain (positive gain/no gain/negative gain). Since Spearman correlation itself is a statistical indicator[1], there is no need to test it again. Then, the prediction accuracy of the indicator is judged. We have updated our results. Prediction accuracy of each indicator: Instance SC (69.6%) and Aggregated SC (89.3%). We will update it in the latest version of the paper.
> [1] The Spearman correlation formula. Science 1905
>
> ### Q2: The result lacks an in-depth theoretical analysis.
> **Response:** Token probability can reflect LLM’s confidence in an answer [1]. Chain-of-thought allows for improving confidence for inherent serial problems by spending more tokens for deep search [2]. However, other not inherently sequential problems might have negative outcomes because of snowball errors [3]. In other words, the model's low confidence may steer it down the wrong reasoning path. When CoT is introduced, the snowball effect is exacerbated, amplifying the initial error further. By incorporating the Spearman correlation indicator, we can assess the model's uncertainty at the start of the reasoning process, helping us determine whether applying CoT is beneficial. Therefore, SC is a good metric to predict the performance gain of CoT. We will make the above content clearer in the latest version of the paper.
> [1] Detecting hallucinations in large language models using semantic entropy. Nature 2024
> [2] Chain of thought empowers transformers to solve inherently serial problems. arXiv 2024
> [3] Rethinking External Slow-Thinking: From Snowball Errors to Probability of Correct Reasoning. arXiv 2025
>
> ### Q3: The paper provides limited discussion on decoding strategies.
> **Response:** Thanks for your suggestion. The decoding strategy is crucial for the output of large models. We conduct experiments on different decoding strategies. We mainly conduct a sensitivity analysis of temperature and top k sampling, and compare the polarity consistency of SC indicators under greedy decoding under different sampling strategies (benchmark granularity). Our conclusion shows that our method remains robust under the change of decoding strategy. For the consistency of different indicators with the greedy decoding strategy after changing the decoding strategy, we obtain Instance_SC (89.7±0.2) and Aggregated_SC (87.3±0.09). We will add a more detailed discussion on decoding strategies in our latest version of the paper. Thank you again for your suggestion.
>
> ### Q4: Writing details in lines 20 and 30.
> **Response:** Thank you for pointing out the writing problem in the paper. We have re-checked the full paper and have corrected the pointed out issues.
>
> ### Q5: The motivation presented in Lines 43–48 is unclear and somewhat contradictory.
> **Response:** Thanks for your question. We clarify our motivation as follows: Judging CoT gain only by task-category is too coarse and cannot explain the inconsistency of task performance within the same category. From a new decoding perspective, we propose a new metric (Instance SC / Aggregated SC) to provide a fine-grained, data-driven effectiveness evaluation criterion for judging CoT gain by quantifying the confidence trend of the model when decoding, which goes beyond empirical inference based on task category. We will update this section in the latest version of the paper.

---

### Decision · Program_Chairs · 2025-05-01

**Decision:**

Accept (poster)

**Comment:**

This paper first presents an observation that the probability of the token predicted increases as more and more tokens are generated. And based on this observation, the authors propose an approach to predict whether CoT would help on a task or not. Experimental results show a very good result.

The main concern from reviewers is that there lacks a theoretical basis for why Token Signature correlates with CoT gain. In the rebuttal, the authors give an intuitive explain about the potential reason behind.